# EXCRETE workflow enables deep proteomics of the microbial extracellular environment
David A. Russo [1,4] ✉, Denys Oliinyk[2,4], Georg Pohnert [1], Florian Meier [2] & Julie A. Z. Zedler [3]

Extracellular proteins play a significant role in shaping microbial communities which, in turn, can impact ecosystem function, human health, and biotechnological processes. Yet, for many ubiquitous microbes, there is limited knowledge regarding the identity and function of secreted proteins. Here, we introduce EXCRETE (enhanced exoproteome characterization by mass spectrometry), a workflow that enables comprehensive description of microbial exoproteomes from minimal starting material. Using cyanobacteria as a case study, we benchmark EXCRETE and show a significant increase over current methods in the identification of extracellular proteins. Subsequently, we show that EXCRETE can be miniaturized and adapted to a 96-well high-throughput format. Application of EXCRETE to cyanobacteria from different habitats (*Synechocystis* sp. PCC 6803, *Synechococcus* sp. PCC 11901, and *Nostoc punctiforme* PCC 73102), and in different cultivation conditions, identified up to 85% of all potentially secreted proteins. Finally, functional analysis reveals that cell envelope maintenance and nutrient acquisition are central functions of the predicted cyanobacterial secretome. Collectively, these findings challenge the general belief that cyanobacteria lack secretory proteins and suggest that multiple functions of the secretome are conserved across freshwater, marine, and terrestrial species.

Bacteria produce and release a remarkable repertoire of small molecules and proteins to communicate and respond to stimuli[1]. Secreted molecules mediate interactions with the environment and the surrounding microbiome, and numerous biotechnological applications take advantage of the secretion machinery. Bacterial protein secretion has mainly been studied in pathogenic and polymer-degrading bacteria due to their importance in human health and the environment[2]. However, so far, protein secretion remains understudied in many environmentally relevant bacteria due to their complex, intractable, habitats where secreted proteins are dilute and difficult to extract. One such example is cyanobacteria.

Cyanobacteria are ubiquitous oxygenic photosynthetic bacteria that contribute at least 10% of global net primary production[3,4], and can be found in freshwater, marine, and hypersaline environments[5–7], as well as in biofilms and microbial mats[8]. The traditional view of a cyanobacterium is that of a self-sufficient photoautotroph that lacks secretory enzymes and is surrounded by a highly impermeable outer membrane that only allows the diffusion of small molecules[9–11]. Challenging this paradigm, emerging evidence suggests that proteins in the cell envelope and extracellular space play key roles in the cyanobacterial lifestyle.

In cyanobacteria, only four out of the eleven bacterial protein secretion systems have so far been identified: type I (T1SS), type IV (T4SS), type V (T5SS), and type IV pili (T4P)[12]. Structurally, cyanobacterial secretion systems resemble their counterparts in other bacteria. Functionally, however, they are involved in remarkably different processes. The T1SS, for example, plays a role in heterocyst development and deposition of the S-layer[13–16]. T4P are involved in phototaxis, DNA uptake, aggregation, and flotation[17–22]. Interestingly, the recent discovery of multiple protein secretion mechanisms that do not involve the aforementioned systems suggests that there may be a myriad of secreted proteins that remain largely unexplored[23–26].

Mass spectrometry (MS)-based proteomics has become the preferred method to study protein secretion and characterize the exoproteome[27]. To date, MS-based exoproteomics studies in cyanobacteria have identified up to a few hundred proteins, with only a fraction of these considered truly secreted[28–30]. This low number is in stark contrast with the abundance of secretory processes already described in cyanobacteria, thus suggesting there is potential for further exploration. To handle the high dilution and sample complexity of the cyanobacterial exoproteome, current workflows for MS-based exoproteomics start with concentration and clean-up steps[21,31,32].

[1]Bioorganic Analytics, Institute for Inorganic and Analytical Chemistry, Friedrich Schiller University Jena, Jena, Germany. [2]Functional Proteomics, Jena University Hospital, Jena, Germany. [3]Synthetic Biology of Photosynthetic Organisms, Matthias Schleiden Institute for Genetics, Bioinformatics and Molecular Botany, Friedrich Schiller University Jena, Jena, Germany. [4]These authors contributed equally: David A. Russo, Denys Oliinyk. ✉e-mail: david.russo@uni-jena.de

Protein precipitation and centrifugal concentrators are widely used. However, these approaches require excessive handling steps and are prone to protein loss. Therefore, the full description of the cyanobacterial exoproteome remains a challenge.

To address this challenge, we developed EXCRETE (enhanced exoproteome characterization by mass spectrometry), an MS-based workflow for direct analysis of microbial exoproteomes adapted from solid-phase enhanced protein aggregation protocols (e.g., SP3[33], protein aggregation capture (PAC)[34]). Utilizing cyanobacteria as a test system, we show that EXCRETE can robustly characterize the exoproteome across species, media, and growth conditions, independent of extracellular matrix, with minimal time and sample handling. Our workflow should be broadly applicable to microbes from a wide range of habitats, with the potential to open new avenues of investigation in microbial exoproteomics.

## Materials and methods
### Cyanobacterial strains and growth conditions
The motile *Synechocystis* sp. PCC 6803 substrain PCC-M (hereafter *Synechocystis* wild-type (WT)) and the *Synechocystis* Δ*hfq* mutant[35] were kindly donated by Annegret Wilde. The PCC-M substrain of *Synechocystis* was chosen because substrains of the non-motile lineage of *Synechocystis* have multiple secretion defects[36]. *Synechococcus* sp. UTEX 3153 (also known as *Synechococcus* sp. PCC 11901) was obtained from the UTEX culture collection. *Nostoc punctiforme* PCC 73102 was kindly donated by Elke Dittmann. *Synechocystis* was maintained on BG-11 medium[37] supplemented with 10 mM 2-tris(hydroxymethyl)-methyl-2-amino 1-ethanesulfonic acid (TES) buffer (pH 8.0) and 1.5% (w/v) Kobe I agar. *Synechococcus* was maintained on AD7 medium[38] supplemented with 1.5% (w/v) Kobe I agar. Both strains were kept at 30 °C with continuous illumination of 40 µmol photons m$^{-1}$ s$^{-1}$. *Nostoc* was maintained in BG-11 medium supplemented with 10 mM TES buffer (pH 8.0) at 22 °C with continuous illumination of 30 µmol photons m$^{-1}$ s$^{-1}$. For the proteomics experiments, *Synechocystis* WT and the Δ*hfq* mutant were grown in 100 mL conical flasks with 20 mL of BG-11 TES medium at 30 °C, continuous illumination of 50 µmol photons m$^{-1}$ s$^{-1}$ and shaking at 150 rpm. The precultures of the Δ*hfq* mutant were supplemented with 15 µg mL$^{-1}$ chloramphenicol. The antibiotic was omitted during experiments to prevent distorting effects. To induce the formation of aggregates, *Synechocystis* was grown in a CellDEG high-density system (HDC 6.10 starter kit; CellDEG)[39]. The CellDEG system was set up as previously described[40], and shaking was set to 100 rpm. The addition of $CO_2$, together with the slow shaking, promotes the aggregating phenotype. *Synechococcus* was grown in BA+ medium (DSMZ ID 1677) in a CellDEG high-density system as previously described[40], with shaking set to 250 rpm. *Nostoc* was grown in T-75 tissue flasks (Sarstedt) with 20 mL of BG-11 TES medium, supplemented with 10 times higher concentration of $Na_2CO_3$, at 21 °C, continuous illumination of 50 µmol photons m$^{-1}$ s$^{-1}$ and no shaking. All pre-cultures were washed twice in their respective media. *Synechococcus* and *Synechocystis* cultures were inoculated to a starting OD$_{750 nm}$ of 0.4 and *Nostoc* cultures to a wet weight of 2.5 g L$^{-1}$. All cultures were grown for 72 h before harvesting for protein extraction. All experiments were conducted on a Unimax 1010 orbital shaker (Heidolph Instruments).

### Preparation of endoproteome and exoproteome fractions
To obtain the exoproteome fraction, cultures were centrifuged for 10 min at 5000×*g*. The supernatant was removed, centrifuged again for 10 min at 10,000×*g* and transferred to a fresh microcentrifuge tube. Samples were kept on ice until processing. Endoproteome fractions were obtained by centrifuging an OD$_{750 nm}$ equivalent of 3 for 10 min at 10,000×*g*. The pellet was then washed and resuspended in 300 µL of lysis buffer (25 mM Tris-HCl, 5% (w/v) glycerol, 1% (v/v) Triton X-100, 1% (w/v) sodium deoxycholate, 0.1% (w/v) sodium dodecyl sulfate (SDS) and 1 mM EDTA). Cells were broken with zirconium oxide beads (diameter 0.15 mm) using a Bullet Blender Storm 24 (Next Advance) with three cycles of 5 min. Cell lysates were then centrifuged at 10,000×*g* for 10 min and the resulting supernatant

transferred to fresh microcentrifuge tubes for protein content determination. Protein content of all samples was determined with a Pierce™ BCA assay kit (Thermo Fisher Scientific).

### Ultrafiltration and in-solution digestion workflow
Exoproteome fractions were concentrated to 1 mg mL$^{-1}$ total protein (approximately 10 times concentration) using an Amicon Ultra-0.5 Centrifugal Filter Unit with a molecular weight cutoff of 3 kDa (Merck Millipore). A 10 µL aliquot of the concentrated exoproteome fraction was reduced with 5 mM tris (2-carboxyethyl) phosphine (TCEP) and alkylated with 5.5 mM chloroacetamide (CAA) at room temperature for 5 min. Subsequently, 90 µL of 25 mM ammonium bicarbonate containing 0.5 µg of MS grade Trypsin/LysC (Promega) (enzyme/protein ratio of 1:20 (w/w)) were added to the sample for in-solution digestion and incubated for 4 h at 37 °C and 1000 rpm. Peptide purification and desalting was performed as described below.

### EXCRETE workflow
Supernatant volumes equivalent to 10 µg (microcentrifuge tube) or 3 µg (microplate) of protein were harvested and transferred to 2 mL microcentrifuge tubes or 96-well microplates, respectively. Sample volumes were normalized to the sample with the highest volume to ensure similar mixing dynamics. NaCl and SDS were added to a final concentration of 10 mM and 1% (w/v), respectively. Samples were then reduced with 5 mM TCEP and alkylated with 5.5 mM CAA at room temperature for 5 min. To induce protein aggregation, LC–MS grade ethanol was added to a final concentration of 50% (v/v) followed by the addition of SiMAG-Carboxyl magnetic particles (product No. 1201, Chemicell) to a final concentration of 0.5 µg µL$^{-1}$. Samples were then incubated for 10 min with shaking at 1000 rpm (750 rpm for microplates) on a Biometra TS1 ThermoShaker (Analytik Jena). Subsequently, magnetic particles were separated on custom made magnetic racks for 60 s. Supernatants were removed and the magnetic particles were washed, on magnet, 3 times with 80% (v/v) ethanol. Washes were discarded. Following the washing steps, samples were removed from the magnetic racks and air dried for 10 min at room temperature to remove residual ethanol. The magnetic particles were then resuspended in 100 µL of 25 mM ammonium bicarbonate containing 0.5 µg of MS grade Trypsin/LysC (Promega) (enzyme/protein ratio of 1:20 (w/w)) for on-bead digestion. Samples were sonicated for 1 min to reconstitute the magnetic particles and incubated overnight (microcentrifuge tubes) or for 4 h (microplate) at 37 °C and 1000 rpm. Following protein digestion, samples were sonicated for 2 min, magnetic particles were separated for 60 s and supernatants were recovered.

### Peptide purification and desalting
Recovered supernatants were diluted into 300 µL wash buffer 1 (1% (v/v) trifluoroacetic acid (TFA) in isopropanol), transferred to SDB-RPS StageTips[41] and centrifuged for 10 min at 1500×*g*. StageTips were washed with 100 µL of wash buffer 1 and, subsequently, with 100 µL of wash buffer 2 (0.2% (v/v) TFA in 5% (v/v) acetonitrile (ACN)) for 8 min at 1500×*g*. Peptides were eluted by adding 60 µL of freshly prepared SDB-RPS elution buffer (0.2% (v/v) NH$_4$OH in 60% (v/v) ACN) and centrifuging for 10 min at 1000×*g*. Eluates were immediately dried under vacuum at 45 °C and stored at −20 °C until analysis. In the microplate workflow, StageTips were processed using a Spin96 device[42]. Before analysis, peptides were resuspended in MS loading buffer (0.1% (v/v) TFA in 2% (v/v) ACN) and the concentration was measured with a NanoDrop Spectrophotometer (Thermo Fisher Scientific) in order to normalize injections to 200 ng of peptides.

### Liquid chromatography–MS analysis
Purified and desalted peptides were separated by nanoflow reversed-phase liquid chromatography in a nanoElute system (Bruker Daltonics) within 60 min at a flow rate of 0.5 µl min$^{-1}$ on a 15 cm × 75 µm column packed, in-house, with 1.9 µm C$_{18}$ beads. Mobile phase A consisted of 0.1% (v/v) formic acid and B of 0.1% (v/v) formic acid in ACN. Peptides, upon elution from the column, were electrosprayed with a CaptiveSpray (Bruker Daltonics) into a

trapped ion mobility spectrometry (TIMS) quadrupole time-of-flight mass spectrometer (timsTOF HT, Bruker Daltonics). Data were acquired in data-independent acquisition with parallel accumulation–serial fragmentation (diaPASEF) mode[43] with an equidistant window scheme in the $m/z$ and ion mobility dimensions. The ion mobility range was set from $1/K_0 = 0.6$ to $1.6\ \mathrm{Vs\ cm^{-2}}$ and equal ion accumulation time and ramp times in the dual TIMS analyzer of 100 ms each. The collision energy was linearly decreased from 59 eV at $1/K_0 = 1.4\ \mathrm{Vs\ cm^{-2}}$ to 20 eV at $1/K_0 = 0.6\ \mathrm{Vs\ cm^{-2}}$. For all experiments the TIMS elution voltages were calibrated by known $1/K_0$ values from at least two out of three ions from Agilent ESI LC/MS tuning mix ($m/z$, $1/K_0$: 622.0289, 0.9848 $\mathrm{Vs\ cm^{-2}}$; 922.0097, 1.1895 $\mathrm{Vs\ cm^{-2}}$; and 1221.9906, 1.3820 $\mathrm{Vs\ cm^{-2}}$).

## Raw data processing

Raw files from diaPASEF were processed in Spectronaut v17[44] (Biognosys) with spectrum libraries built directly from the DIA experiments with the directDIA+ workflow. False discovery rates were set to <1% at peptide and protein levels using a target-decoy based approach. Cysteine carbamido-methylation was set as a fixed modification and protein N-terminal acet-ylation and methionine oxidation were set as variable modifications. All spectra were matched against *Synechocystis* (GenBank: CP003265.1), *Synechococcus* (UniProt: UP000304321) and *Nostoc* (UniProt: UP000001191) reference proteomes (accessed September 2022). Quantifi-cation values were filtered by $q$-value and the 'Automatic' normalization mode was defined for cross run normalization, while the imputation strategy was set to 'None'. Report data matrices were exported on protein group level for downstream bioinformatic analysis.

## Bioinformatic analysis

Proteins were annotated using EggNOG v5.0[45], PsortB v3.0[46], SignalP 6.0[47] and UniProtKB[48]. Proteins were considered secreted when PsortB or Uni-ProtKB predicted the location as "Periplasm", "Outer Membrane", or "Extracellular". In addition, proteins with a location predicted as "Unknown" were considered secreted when a signal peptide was detected by SignalP. Reciprocal best hits between the same COG categories across species were detected and scored with BLAST using the Galaxy EU web platform (https://usegalaxy.eu/)[49,50]. Proteins were considered orthologs if detected with a query cover >50% and a bit score >50. Data analysis and visualization were performed using custom scripts in R (4.0.1) and Python (3.9.4) with packages data.table (1.14.2), dplyr (1.0.7), ggplot2 (3.3.5), tidyR (1.1.14), pandas (1.1.15), numpy (1.22.2), plotly (5.4.0), scipy (1.7.3).

## Statistics and reproducibility

All experiments were performed with three to five replicates. Data are presented as the mean and individual data points are plotted unless stated otherwise. Detailed information is given in the respective figure legend. A protein group was considered identified when it was present in at least 70% of the replicates with a minimum of three replicates. Missing values were then imputed using the k-nearest neighbors algorithm. In Fig. 1d and Fig. 2a, the number of protein groups identified is given before filtering and imputation. Differential protein analysis in Fig. 5d was done using a Stu-dent's two-sample unpaired $t$-test with permutation-based multiple test correction. Differentially produced proteins were selected from all identified proteins with a cutoff criterion of fold change = 2 and adjusted $p$-value < 0.05.

## Reporting summary

Further information on research design is available in the Nature Portfolio Reporting Summary linked to this article.

## Results

### Design of a solid-phase enhanced protein aggregation workflow for cyanobacterial exoproteomics

Exoproteomic samples from cyanobacteria typically suffer from high dilu-tion and the presence of substantial amounts of other compounds such as

inorganic salts, lipids, and polysaccharides. To date, these challenges have been approached with labor-intensive concentration methods such as protein precipitation, ultrafiltration, and dialysis[21,30–32,51]. These techniques are time-consuming, prone to loss of material, and, ultimately, lead to unsatisfactory results (Fig. 1a). Therefore, we sought to develop a fast, robust, and unbiased workflow that allows easy exoproteome concentration and subsequent removal of contaminants. To this end, we took inspiration from solid-phase enhanced protein aggregation protocols (e.g., SP3 and PAC) that have already been applied to exoproteomic samples, albeit of human model systems[34,52]. These protocols rely on a phenomenon where precipitated and aggregated proteins bind to a solid phase irrespective of its surface chemistry. Paramagnetic beads are the most widely used solid phase. However, a variety of other microparticles, including glass beads, have also been successfully applied[33,34,53,54]. Protein aggregation and binding is con-sidered to be unbiased and can be induced with a variety of solvents (e.g., acetonitrile, ethanol) and conditions (e.g., high temperature and high salt)[34,53]. Solid-phase enhanced protein aggregation workflows are scalable and can be performed in less than 1 h in a single container, thus allowing for minimal sample loss and same-day analysis (Fig. 1a). In our con-ceptualization of these protocols, termed EXCRETE (Fig. 1b), chlor-oacetamide (CAA) and tris(2-carboxyethyl)phosphine (TCEP) are used to allow the alkylation and reduction steps to be combined with the protein aggregation step (Fig. 1b). Then, we chose carboxylate-modified para-magnetic beads as an economic solid-phase that can be added at a high concentration ($0.5\ \mathrm{\mu g\ mL^{-1}}$) to facilitate the capture of proteins in typically dilute extracellular samples[34]. Protein aggregation is induced by adding ethanol to a final concentration of 50% (v/v). Importantly, the paramagnetic beads can then be extensively washed with 80% ethanol (v/v) to remove co-aggregating contaminants (Fig. 1b). Protein digestion is then performed on-bead in as little as 4 h. Following digestion, peptides are recovered by col-lecting the supernatant following bead collection on a magnetic rack.

To benchmark our bead-based workflow, we compared the perfor-mance of EXCRETE against an ultrafiltration-based method, as reported previously[16,55]. For the comparison, we chose exoproteomic samples of the marine cyanobacterium *Synechococcus* sp. UTEX 3154 (hereafter *Syne-chococcus*). *Synechococcus* is a recently isolated unicellular marine cyano-bacterium of biotechnological interest due to its fast, sustained, biomass accumulation[56]. This also allowed us to test how EXCRETE performs on seawater samples.

Regarding the number of identified peptides and proteins, EXCRETE clearly outperformed ultrafiltration. On average, 3974 peptides and 639 proteins were identified using EXCRETE (Fig. 1c, d, Supplementary Data 1). In the case of ultrafiltration, we identified, on average, 2549 peptides and 339 proteins (Fig. 1c, d, Supplementary Data 2). On-bead protein digestion has been shown to reduce the number of missed tryptic cleavages in comparison to in-solution digestion[34], therefore we investigated whether this was also the case in our benchmarking experiment. EXCRETE and ultrafiltration fol-lowed by in-solution digestion showed similar number of missed cleavages (Supplementary Fig. 1a). We then proceeded to assess the precision of each workflow by calculating the coefficient of variation (CV) of the raw protein intensity amongst the replicates and found a median CV of 14% for the ultrafiltration workflow and 27% for EXCRETE (Supplementary Fig. 1b, c). This is in line with previous reports showing that bead-based methods can present higher CVs[57]. Finally, regarding protein identifications, we observed an overlap of 47% of proteins with 49% only identified with EXCRETE and 4.1% exclusive to the ultrafiltration workflow (Fig. 1e, Supplementary Data 1, Supplementary Data 2). The additional proteins identified included several low abundance proteins (median rank of unique proteins = 348), thus demonstrating the robustness of our method across a wide dynamic range (Supplementary Fig. 1b, Supplementary Data 3).

### Miniaturization of the EXCRETE workflow allows for increased throughput without loss of information

Having successfully benchmarked EXCRETE, we aimed to miniaturize the workflow to allow for high-throughput analysis in 96-well microplates. To

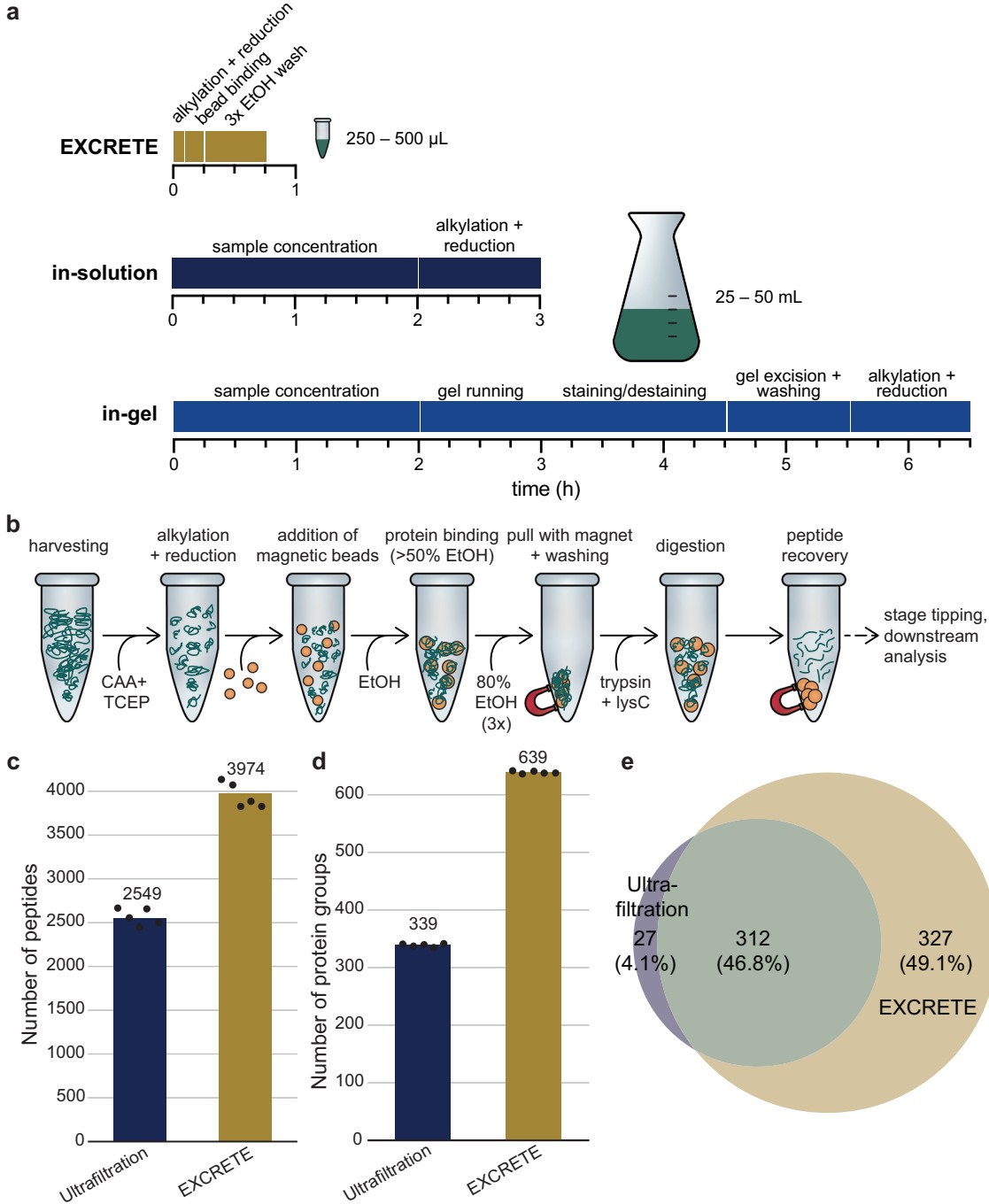

**Fig. 1 | EXCRETE outperforms traditional workflows for exoproteomic sample preparation. a** Comparison of workflows, time required and input volume between the classic in-gel and in-solution approaches and EXCRETE. **b** Detailed EXCRETE sample preparation workflow. **c, d** Number of peptides (**c**) and protein groups (**d**) identified by ultrafiltration and EXCRETE (black dots represent biological replicates, $n = 5$). **e** Venn diagram representing the overlap of protein identifications with ultrafiltration and EXCRETE.

adapt to a microplate format, we decreased all initial volumes of culture supernatant to a maximum of 150 µL, consequentially decreasing the starting protein amount threefold from ~10 to ~3 µg. To evaluate the efficiency of our miniaturized high-throughput workflow in comparison with its microtube-based analog, we analyzed the *Synechococcus* exoproteome with both approaches. Regarding protein identification, both approaches were quantitatively similar with $272 \pm 40$ and $273 \pm 7$ protein groups identified after filtering, using the microplate and microtube workflows, respectively (Fig. 2a, Supplementary Data 4, 5). Our data revealed good quantitative reproducibility with median CV of 24% in the microtube workflow and 22% for the microplate workflow (Fig. 2b). In addition, the

microtube workflow covered a marginally higher fraction of low abundance proteins (Fig. 2b). Quantitative precision was high for both approaches with a Pearson correlation coefficient of $r = 0.94$ within replicates processed with microplates (Fig. 2c) and $r = 0.91$ between replicates processed with microplates and microtubes (Fig. 2d). We then sought to compare the accuracy of protein identification between the microplate and microtube workflows. More than 75% of total proteins identified were found using both approaches (Fig. 2e). The unique proteins identified in each workflow were generally of lower abundance (median rank of unique proteins = 215 in microtubes and 201 in microplates) (Fig. 2f, Supplementary Data 6 and 7). Finally, to evaluate how the reproducibility of the microplate-workflow

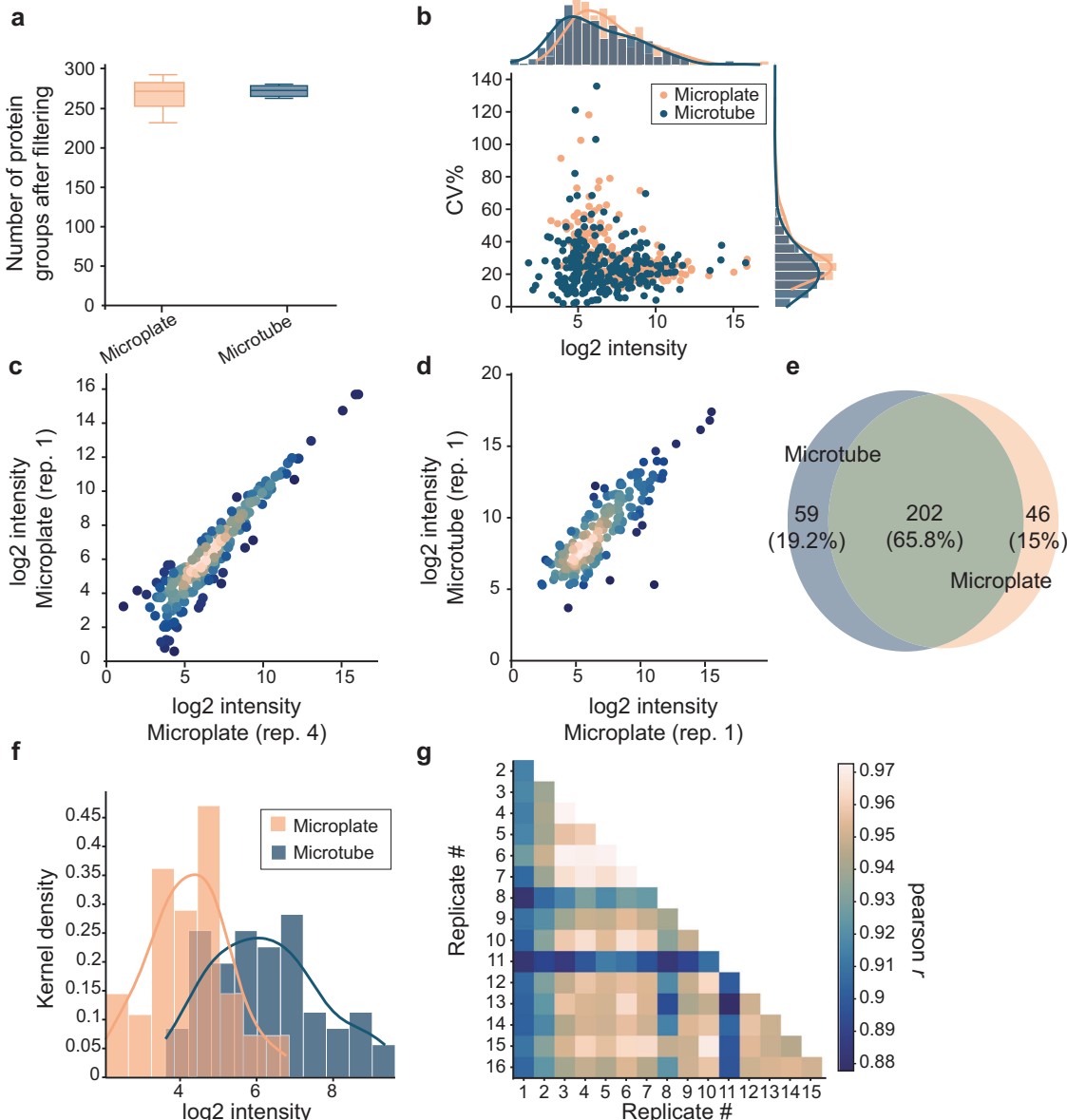

**Fig. 2 | EXCRETE can be miniaturized to a microplate format without loss of information. a** Number of protein groups identified by microtube- and microplate-based workflows following filtering and imputation. Center line of boxplots, median; box limits, upper and lower quartiles; whiskers, minimum to maximum values; $n = 3$ biological replicates. **b** Coefficients of variation (CVs) of the raw protein intensities ordered by log2 protein intensity. On the secondary $x$-axis histogram and density plots representing the frequency distribution of protein intensities are shown. On the secondary $y$-axis histogram and density plots representing the frequency distribution of CVs are shown. Dots represent means of biological replicates. **c, d** Correlation of intensities of identified proteins in between biological replicates processed with (**c**) the microtube- based method or (**d**) the microtube- and microplate-based methods. **e** Venn diagram representing the overlap of protein identifications with microtubes and microplates. **f** Intensity distribution of unique proteins identified with microtubes and microplates. **g** Pairwise Pearson correlations of sixteen microplate workflow replicates.

scales towards higher throughput, we measured sixteen replicates processed in parallel with the same workflow. The pairwise Pearson correlation coefficient was higher than 0.88 across all injections (Fig. 2g). Together, these results show that, despite a lower protein input, the microplate workflow is qualitatively and quantitatively equivalent to the microtube-based workflow. Miniaturization of EXCRETE will allow for the analysis of dozens of secretomes per day and opens the door to automation of exoproteome analysis on robotic liquid handling platforms[58].

**EXCRETE allows for deep exoproteomic profiling of cyanobacteria from a range of habitats**

Solid-phase enhanced protein aggregation protocols have already been successfully applied to analyze the exoproteome of human tissue

cultures[34,52]. However, in contrast to these, bacteria grow in a wide range of environments that can be rich in salt, lipids, and polysaccharides. Given that EXCRETE performed well with seawater exoproteome samples from a marine cyanobacterium, we decided to test our workflow on another two cyanobacteria with different extracellular matrices. To this end, we compared the performance of EXCRETE on three cyanobacteria from distinct habitats: the aforementioned marine *Synechococcus*; *Synechocystis* sp. PCC 6803 (hereafter *Synechocystis*), a motile unicellular freshwater cyanobacterium; and *Nostoc punctiforme* PCC 73102 (hereafter *Nostoc*), a terrestrial $N_2$-fixing filamentous cyanobacterium which retains the developmental complexity of field isolates and is a model for plant-cyanobacteria symbiosis[59,60]. *Synechocystis* and *Nostoc* are also known to secrete large amounts of polysaccharides[60,61], thus allowing us

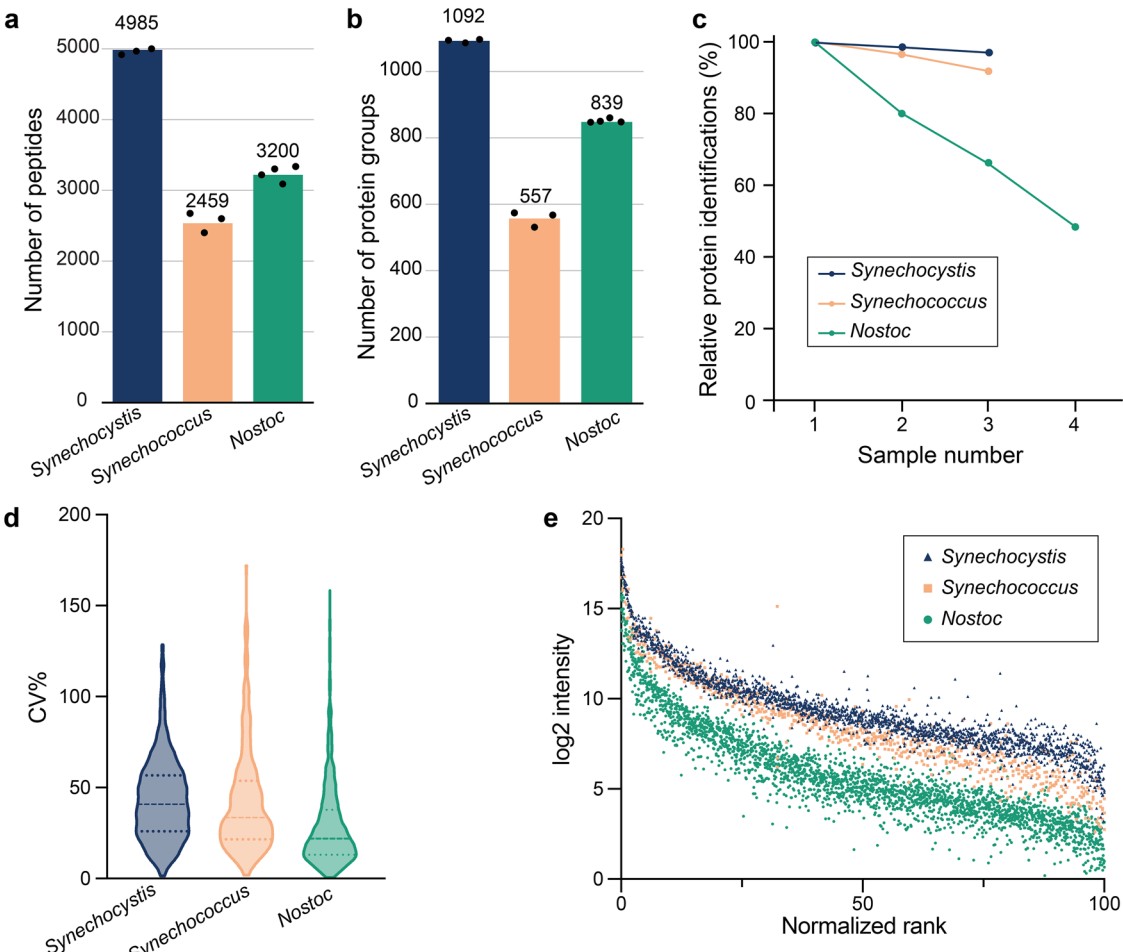

**Fig. 3 | The EXCRETE workflow enables robust and direct sampling of exo-proteomes from *Synechocystis*, *Synechococcus* and *Nostoc* cultures. a, b** Number of peptides (**a**) and protein groups (**b**) identified in the exoproteome of each species. Means are shown above the bars. Black dots represent biological replicates (*n* = 4 for *Synechocystis* and *Nostoc*, *n* = 3 for *Synechococcus*). **c** Percentage of protein groups present with an increasing number of replicates for each species. **d** Coefficient of variation (CV) of the raw intensities of proteins identified across all biological replicates of each species. Dashed line represents the median. Dotted lines represent the top and bottom quartiles. **e** Log2 protein intensities ordered by rank. Dots represent individual protein replicates. Protein ranks have been normalized to a range between 0 and 100.

to test EXCRETE on polysaccharide-rich matrices. Analysis of 10 μg exoproteomic samples (approximately 600 μL of harvested supernatant for *Synechocystis* and 300 μL for *Synechococcus* and *Nostoc*) led to the identification of 4985, 2459, and 3200 peptides and 1092, 557, and 839 protein groups, at 1% false discovery rate (FDR) at both peptide and protein level, in *Synechocystis*, *Synechococcus*, and *Nostoc*, respectively (Fig. 3a, b). We observed that 80% data completeness (i.e., values present in 80% of replicates) was achieved for 97.5%, 91.2%, and 62.8% of all protein groups identified in *Synechocystis*, *Synechococcus*, and *Nostoc*, respectively (Fig. 3c). The raw data was then filtered (only protein groups present in at least 70% of the replicates with a minimum of three replicates were retained) and imputed using the k-nearest neighbors algorithm. Following data filtering and imputation, 885, 261 and 668 proteins, in *Synechocystis*, *Synechococcus*, and *Nostoc*, respectively, were kept for further analysis (Supplementary Data 4, 8, 9). Compared to previous studies using TCA precipitation[29,30], this is a 7.5-fold increase for *Synechocystis*, a 2.3-fold increase for *Nostoc*, and the first report for this *Synechococcus* species (Supplementary Fig. 2a, b). All three species showed good biological reproducibility with median CVs of 43%, 24% and 22% for *Synechocystis*, *Synechococcus*, and *Nostoc*, respectively, and a general trend of lower dispersal at higher intensities (Fig. 3d, e). Altogether, these results demonstrate that our approach is robust and performs well for all species and extracellular matrices tested here.

## Multiple functions of the predicted cyanobacterial secretome are conserved across species

Besides secretory routes, proteins can be released to the extracellular milieu through unspecific mechanisms such as cell lysis, defective cell division, and extracellular vesicles[62–64]. Therefore, having identified a large number of exoproteins in this work, we sought to identify which proteins were actively translocated beyond the plasma membrane. To define this subset of the exoproteome of each cyanobacterium we combined PsortB[46], SignalP[47] and UniprotKB[48] for the prediction of subcellular location and signal peptide recognition. Regarding location, historically, a secreted protein was defined as one exclusively transported via a secretion system[65]. However, this fails to encompass the known complexity of the Gram-negative cell envelope where a protein can be active in both the periplasm and the extracellular space and can be secreted via non-canonical secretion systems. Therefore, we have adopted a less stringent definition of secretome encompassing proteins predicted to be translocated to the periplasm, outer membrane or extra-cellular milieu[66]. Regarding signal peptides, in bacteria the presence of a signal peptide typically suggests a protein is secreted. However, some secretion pathways (e.g., T1SS) do not utilize classical N-terminal signal peptides, therefore the prediction of a signal peptide was not required for a protein to be considered secreted. In addition, signal peptides directing proteins to the thylakoid membrane cannot be distinguished from those directing proteins to the exterior of the cell[12,67]. Therefore, proteins predicted

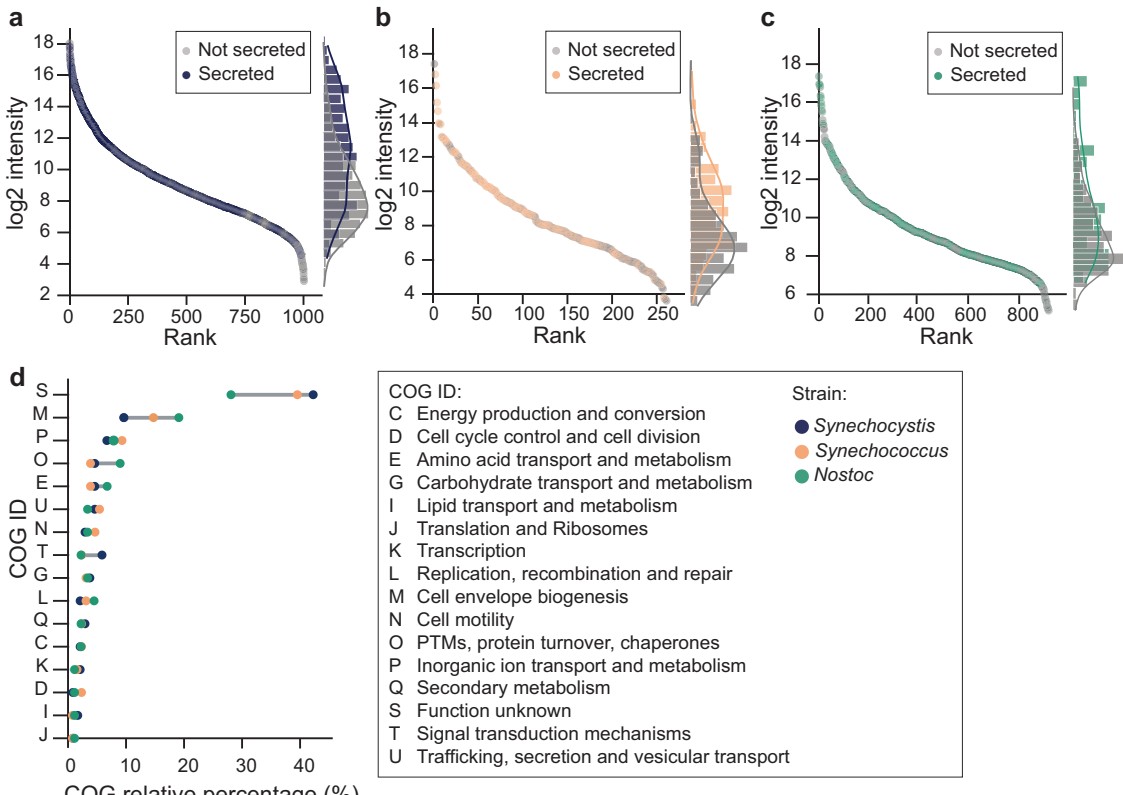

**Fig. 4 | The secretome functional profile is conserved across *Synechocystis*, *Synechococcus*, and *Nostoc*. a–c** Protein intensities ordered by rank with indication of secreted (colored dots) and non-secreted (gray dots) proteins in (**a**) *Synechocystis*, (**b**) *Synechococcus*, and (**c**) *Nostoc*. Dots represent means of biological replicates. On the secondary y-axis histogram and density plots representing the frequency distribution of secreted and non-secreted proteins are shown. **d** Clusters of Orthologous Genes (COG) analysis of secreted proteins in each species. COG relative percentage represents the number of times a COG ID appears in relation to the total numbers of COG IDs.

to have signal peptides, but also predicted to be localized in the thylakoid membrane, were removed from the final set of secreted proteins.

Following the criteria defined above, we detected 156, 117, and 81 proteins predicted to be secreted in *Synechocystis*, *Synechococcus*, and *Nostoc* (corresponding to 62%, 46% and 14% of all potentially secreted proteins in the respective proteomes) (Supplementary Data 4, 8, and 9). Applying the same criteria to published datasets showed that, in *Synechocystis*, only up to 8% of potentially secreted proteins have been identified using ultrafiltration-based sample preparation (Supplementary Fig. 2c)[16]. In *Nostoc*, we found a better overlap with a study using precipitation-based sample preparation (13% of potentially secreted proteins (76 proteins)[30] vs 14% (81 proteins) in this study (Supplementary Fig. 2d)). The identification of only 14% of the potential *Nostoc* secretome, in comparison to 62% in *Synechocystis*, is curious due to the fact that *Nostoc* encodes a much larger number of secretion systems than *Synechocystis*[12]. This is in line with previous work showing that multicellular symbiotic cyanobacteria, such as *Nostoc*, exhibit high levels of proteomic plasticity and only express a small fraction of their proteome at any given point in time[30,68–70]. Accordingly, our microscopic observations showed that the *Nostoc* culture was mostly in a vegetative state with only a few heterocysts (Supplementary Fig. 3). Therefore, the absence of different cell types might also explain the low number of protein identifications.

To determine whether the proteins we predicted as secreted were present at higher intensities in the exoproteome, all proteins identified were plotted by rank and intensity. In support of our experimental approach, secreted proteins clustered at the top of the curve (Fig. 4a–c) and accounted for 56%, 60%, and 48% of the total exoprotein intensity in *Synechocystis*, *Synechococcus*, and *Nostoc*, respectively (Supplementary Fig. 4). In addition, to investigate to what extent proteins identified outside the cell stem from unspecific processes such as cell lysis, we evaluated the relative abundance of

intracellular marker proteins (i.e., photosystem subunits, phycobilins, ribosomes, and RuBisCO) in the detected exoproteomes of the three cyanobacterial species. These proteins accounted for 9%, 2%, and 10% of the total protein intensity in *Synechocystis*, *Synechococcus*, and *Nostoc*, respectively (Supplementary Table 1). In comparison, the intensity of these proteins accounted for 38% and 32% of the *Synechocystis* and *Nostoc* endoproteomes, respectively (Supplementary Data 10 and 11). These numbers compare favorably to previous studies of marine *Synechococcus* species where these intracellular markers accounted for 14 – 50% of the respective exoproteomes (Supplementary Table 1)[71,72].

We then proceeded to investigate which proteins were identified in the predicted secretomes to confirm whether the patterns observed had support in the literature. First, we observed that approximately 78% of all proteins predicted as secreted contain a Sec signal peptide. This is in line with what is known in cyanobacteria where the majority of proteins are translocated via Sec[12]. Next, PilA1, the major component of the retractable fiber of the T4P system, was the most abundant secreted protein in *Synechococcus* (UniProtKB: A0A4P8WZ44), fourth most abundant in *Synechocystis* (UniProtKB: P73704) and was not identified in *Nostoc*. This was expected as cell surface appendages are amongst the most prominent proteins typically found in the secretomes of unicellular cyanobacteria[36]. Despite *Nostoc* possessing a functional T4P system, PilA1 is only expressed in hormogonia[60,73]. Given hormogonia were not found during microscopic inspection of the cultures, the absence of PilA1 agrees with the observed lack of differentiated cells (Supplementary Fig. 3). Other examples of cell surface appendages found include PilX1 (*Synechocystis*, UniProtKB: P73704), where a potential role in flocculation has been proposed[74], and CccS (*Synechocystis*, UniProtKB: P74672), involved in the construction of cell surface components[75]. The cyanobacterial secretome has also been

implicated in microelement acquisition and detoxification[76]. Our data also strongly support this role. The second most abundant proteins predicted as secreted in *Synechocystis*, *Synechococcus* and *Nostoc* are FutA2 (UniProtKB: Q55835), a periplasmic iron-binding protein[77], PhoX (UniProtKB: A0A4P8X4T0), a secreted alkaline phosphatase[78], and a phosphonate ABC transporter (UniProtKB: B2J991), respectively. Porins harboring SLH (S-layer homology) and OprB (carbohydrate-selective porin OprB) protein domains, which have also been connected to OM transport functions, were also identified in all three species. Finally, it is also noteworthy that we identified several large (100–490 kDa) proteins in the predicted secretome of *Synechocystis* (Supplementary Table 2). These proteins were generally characterized by the absence of signal peptides, the presence of $Ca^{2+}$ binding motifs, pI values below pH 5 and an amino acid composition rich in glycine residues but with only a few, if any, cysteines. Taken together, this suggests that these proteins are T1SS substrates[79]. Large secreted proteins, reaching up to 2.72 MDa, have also previously been identified in multiple marine and freshwater cyanobacteria[80,81].

Next, to explore the functional aspects of the individual species, we annotated the subset of proteins predicted as secreted with Clusters of Orthologous Genes (COG) categories. We found that in *Synechocystis*, *Synechococcus*, and *Nostoc*, 20.6%, 0%, and 5.6% of all proteins predicted as secreted lacked a match to the COG database and a further 22.8%, 39.5%, and 22.5% were annotated as proteins of unknown function. Amongst the proteins with known COG functions, all three predicted secretomes showed similar functional profiles (Fig. 4d); the most highly represented COG functions were cell wall/membrane/envelope biogenesis (9–19%), inorganic ion transport and metabolism (3–9%), and posttranslational modification, protein turnover, chaperones (4–9%) (Fig. 4d). Amongst all the categories, signal transduction mechanisms showed the largest relative deviation, with this category being represented almost four times higher in *Synechocystis* than in *Synechococcus* and *Nostoc*. In absolute terms, the largest deviation was seen for cell wall/membrane/envelope biogenesis with a difference of nearly 10% between species.

Given that the functional profile was conserved across all three species, we proceeded to investigate the degree of conservation between the proteins predicted as secreted within each of the top COG categories. To this end, we used BLAST to map the reciprocal best hits between each category from each species and the same category in the other two species. The results showed that the posttranslational modification, protein turnover, chaperones, and cell wall/membrane/envelope biogenesis were the most conserved categories with an average of 45% and 55%, respectively, of proteins having orthologs in at least one other species. The categories unknown function and inorganic ion transport and metabolism were less conserved with only 28% and 23%, respectively, of proteins having orthologs in at least one other species (Supplementary Table 3). Altogether, these results suggest that cell envelope maintenance and nutrient acquisition are core functions of the predicted secretomes albeit with different levels of ortholog conservation across cyanobacteria from different habitats.

## The stable *Synechocystis* secretome is enriched in proteins involved in cell envelope maintenance

Bacterial secretomes are remarkably dynamic and have been shown to respond to different media and growth conditions[31,76,82]. The proportion of the secretome that does not vary across conditions can be considered "stable" and is likely constituted by proteins that perform essential services. Therefore, we analyzed the predicted *Synechocystis* secretome in different conditions to determine whether a stable set of proteins would emerge.

To this end, we compared the predicted *Synechocystis* secretome in the WT condition as described above (Fig. 3, Supplementary Data 8, Supplementary Fig. 5a) with that of a bloom condition and a Δ*hfq* mutant (Fig. 5a). In the bloom condition *Synechocystis* was cultivated in high-$CO_2$ to form aggregates embedded in a viscous EPS-rich extracellular matrix (Supplementary Fig. 5b). In this condition we hypothesized that, together with EPS secretion we would see an increase in protein secretion and an expanded predicted secretome. In the Δ*hfq* condition we used a *Synechocystis* Hfq

knockout mutant. Current evidence suggests that the cyanobacterial Hfq binds to the T4P extension motor PilB1 and regulates T4P biogenesis. The loss of Hfq, and consequent loss of T4P function, abrogates several processes that depend on pilin proteins such as motility, natural competence and aggregation (Supplementary Fig. 5c)[20,32,35,83]. In addition, this mutant also exhibited a 73% decrease in the secretion of a nanoluciferase reporter[24]. Therefore, in the Δ*hfq* condition we expected a reduced predicted secretome. Accordingly, we identified 2005 and 397 proteins in the exoproteomes of the bloom condition and the Δ*hfq* condition, respectively (Supplementary Fig. 6). Of these, 212 and 127 were predicted as secreted (Supplementary Fig. 6, Supplementary Data 12 and 13). In comparison to the WT condition, this represents an increase of 56 proteins in the bloom condition and a decrease of 29 proteins in the Δ*hfq* condition.

Looking at the individual conditions, when forming bloom-like aggregates, *Synechocystis* exhibited an expanded predicted secretome, covering nearly 85% of potentially secreted proteins, with 55 proteins unique to this condition (Fig. 5b). Previous work suggests that EPS-rich extracellular matrices can sequester small molecules and nutrients, thus acting as an extracellular pool of resources available for mobilization by the cyanobacterial secretome[84,85]. We thus reasoned that the expansion of the predicted secretome could be linked to the mobilization of extracellular resources. To investigate this hypothesis, we analyzed which COG categories were enriched in comparison to the predicted secretome shared by all three conditions. In agreement with our hypothesis, we observed a combined 26% increase in the inorganic ion transport, energy production, and amino acid transport categories (Fig. 5c). These findings support the idea that cyanobacteria are not only major contributors to the extracellular matrix but can also utilize it as a nutrient resource.

In the Δ*hfq* condition, *Synechocystis* exhibited a reduced predicted secretome with only 127 proteins predicted as secreted compared to 156 in the WT condition. Furthermore, 30 proteins, identified in the WT condition, were absent from the predicted secretome of the Δ*hfq* mutant (Fig. 5b, unique proteins listed in Supplementary Data 14). We, therefore, wondered whether the proteins lacking from the Δ*hfq* mutant were entirely absent from the cellular proteome or simply not secreted, i.e., present and accumulating in the cell. To test this, EXCRETE was applied to examine the endoproteome of the the Δ*hfq* mutant and compare it to the WT endoproteome. 1905 and 2176 proteins were identified in the WT and Δ*hfq* mutant endoproteomes, respectively (Supplementary Data 10 and 15). Amongst the endoproteins in the Δ*hfq* mutant, approximately half of the 30 proteins absent from its secretome could be detected with no significant difference in abundance compared to the WT (Supplementary Fig. 7). The lack of secretion, therefore, did not result in intracellular accumulation. Finally, we observed that out of 56 differentially regulated proteins between the Δ*hfq* and the WT conditions, 54 were upregulated in the Δ*hfq* condition (Fig. 5d). Looking at the COG categories of these 54 proteins, functions in cell envelope assembly and maintenance were most prominent (19%) (Supplementary Table 4). It has previously been suggested that loss of Hfq has a pleiotropic effect and the Δ*hfq* mutant exhibits a compromised cell envelope[24]. Therefore, an upregulation of proteins involved in cell envelope maintenance may constitute a response to cell envelope stress.

Finally, we sought to identify the function of the stable set of proteins predicted as secreted in *Synechocystis*. A total of 126 proteins were identified across all three conditions (Fig. 5b). Interestingly, the predicted secretome in the Δ*hfq* condition almost completely overlapped with the stable secretome, which suggests the loss of Hfq reduced the secretome to essential functions. Among the stable 126 proteins predicted as secreted, the most represented COG category with known function was cell envelope assembly and maintenance (11%) (Fig. 5e). Furthermore, this was the only category enriched in the stable subset of proteins with an 8% increase in comparison to the variable subset of proteins (Fig. 5e). This supports our conclusion that the core function of the cyanobacterial secretome is to assemble and maintain critical external barriers that protect their photoautotrophic lifestyle.

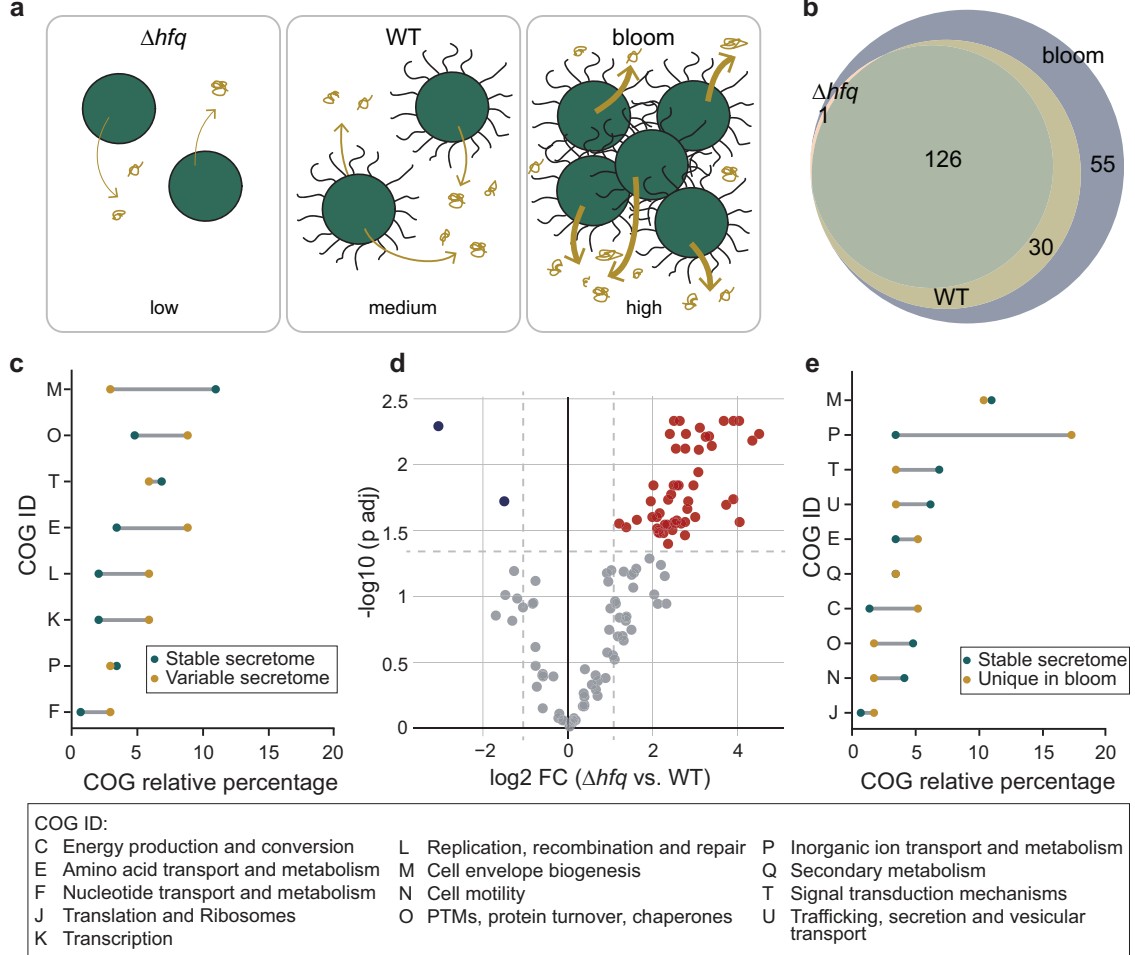

**Fig. 5 | Secretome comparison between wild-type (WT) *Synechocystis*, a Δ*hfq* mutant and a bloom-like culture. a** Schematic overview of *Synechocystis* secretome comparison resulting in low (Δ*hfq*), medium (WT) and high (bloom-like culture) levels of protein secretion. **b** Venn diagram representing the overlap of protein identifications. **c** Comparison of Clusters of Orthologous Genes (COG) category distribution between all secreted proteins in the stable secretome and those unique to bloom conditions. **d** Volcano plot illustrating differential protein expression between proteins identified in the secretomes of *Synechocystis* in the WT and Δ*hfq* conditions. Each dot on the plot represents an individual protein (identification derived from $n = 4$ biological replicates). A Student's two-sample unpaired $t$-test with permutation-based multiple test correction was used for differential proteins analysis. The vertical and horizontal dashed lines represent the fold-change (plotted at −1 and 1) and significance (plotted at 0.05) thresholds, respectively. Color coding indicates significant changes (red for upregulated, blue for downregulated, and gray for non-significant changes). **e** Comparison of COG category distribution between all secreted proteins in the stable secretome and those in the variable secretome. COG relative percentage represents the number of times a COG ID appears in relation to the total numbers of COG IDs.

## Discussion

Cyanobacteria are pioneer organisms that play a key role in shaping global habitats. Therefore, description of the cyanobacterial exoproteome is crucial to understand the mechanisms through which cyanobacteria shape their environment and associated microbiome. In this study we developed EXCRETE, a workflow based on solid-phase enhanced protein aggregation protocols that enabled deep investigation of the cyanobacterial exoproteome. Current methods to interrogate the exoproteome typically start with sample concentration by ultrafiltration or protein precipitation and are often followed by additional clean-up steps. Here, protein losses can occur due to adsorption to the ultrafiltration column, difficulties in washing and resolubilizing the protein pellet following precipitation, and sample degradation due to long processing times and excessive handling. EXCRETE overcomes these issues by concentrating proteins on paramagnetic beads that can be extensively washed with little to no loss of material. This is done in a single container, does not require specialized equipment and samples can be prepared in less than 1 h (Fig. 1a, b). The superior performance of EXCRETE was demonstrated through benchmarking against an ultra-filtration workflow. With EXCRETE we identified 88% more protein groups resulting in 327 unique protein identifications (Fig. 1d, e). Interestingly,

these unique identifications included many low abundance proteins, thus increasing the dynamic range (Supplementary Fig. 1b, Supplementary Data 3). One aspect where we did not see an improvement was in protein digestion efficiency (Supplementary Fig. 1a). However, this is likely due to the use of a relatively high 1:20 enzyme-to-protein ratio. Previous studies have shown that protein digestion on paramagnetic beads is robust down to a 1:50–1:1000 enzyme-to-protein ratio[34], therefore this could be an area of improvement for future iterations of EXCRETE. We also demonstrated that EXCRETE could be adapted to a 96-well microplate format (Fig. 2). Miniaturization of the workflow will allow processing of hundreds of samples per day, opening the door to large-scale exoproteomic studies. Solid-phase enhanced protein aggregation workflows have already been automated[58]. Therefore, we expect that our workflow could easily be transferred to a liquid handling robot for high-throughput exoproteome analysis.

In comparison to previous studies, application of EXCRETE to three different cyanobacteria enabled the detection of a significantly higher number of proteins predicted to be secreted (Supplementary Fig. 2c, d). However, the identification of such a large number of exoproteins was somewhat unexpected given that cyanobacteria are not typically considered proficient protein secreters[11]. Cells are surrounded by a thick, multilayered

envelope with a considerably lower permeability than typical Gram-negative bacteria and porins that only allow the diffusion of inorganic solutes[10]. A COG analysis of secreted proteins pointed towards a specialization in cell envelope management (which includes protein turnover), and nutrient acquisition. All of which, interestingly, were conserved across all species and conditions tested (Figs. 4 and 5). These results are not at odds with the traditional view of cyanobacteria. With the ability to synthesize organic compounds themselves, cyanobacteria do not need exoproteins to hydrolyze organic matter but rather to maintain their protective exterior and facilitate the acquisition of nutrients essential to sustain their photoautotrophic lifestyle[9].

A closer look at the individual proteins across species showed that multiple protein orthologs present in all three species were identified in the categories of protein turnover and cell wall/membrane/envelope biogenesis (Supplementary Table 5). In the former category, the widely conserved Deg/HtrA endopeptidases HhoA and HhoB were identified. In the latter category, porins (e.g., with SLH/OprB-domains) and enzymes such as N-acetylmuramoyl-L-alanine amidases, lytic transglycosylases, and peptidyl-prolyl cis-trans isomerases were identified in all species. A recent study in *Anabaena* sp. PCC 7120 also identified several extracellular amidases with potential roles in peptidoglycan recycling and the formation of nanopores[51]. This suggests that enzymes involved in cell envelope assembly and maintenance are ubiquitous and are a promising direction of future studies in cyanobacteria. The categories of nutrient acquisition and unknown function contained less orthologs shared in all three species. In the former category, this is a result of different nutrient preferences. *Synechococcus* showed a specialization in Fe metabolism while the *Synechocystis* and *Nostoc* exoproteomes contained more proteins related to P, S, and Zn metabolism. In the latter category, we could not determine the function of the shared orthologs but the beta lactamase fold, pentapeptide repeats and tetratricopeptide repeat motifs were present in all three species.

In this study we have greatly expanded the number of proteins predicted to be secreted in cyanobacteria. However, it remains unclear how hundreds of secreted proteins are transported across a tightly regulated cell envelope. For example, our data support the presence and activity of the one-step T1SS. However, putative T1SS substrates only accounted for a small fraction of total secreted proteins (Supplementary Table 2). For two-step secretion in cyanobacteria, it is generally accepted that translocation across the cytoplasmic membrane is mediated by the Sec and Tat pathways[12,67]. Less is known regarding translocation across the outer membrane (OM). In bacteria, the type II secretion system (T2SS) mediates the secretion of a wide variety of proteins. However, cyanobacteria only possess the homologous T4P system which, some have speculated, could act as a T2SS[32]. Our previous work has shown that inactivation of the T4P OM pore, PilQ, did not change the secretion levels of a heterologous reporter. Thus, suggesting that the T4P system is likely dedicated to the secretion of pilins[24]. In support of this, PilA1 is significantly less abundant (59-fold) in the secretome of the *Synechocystis* Hfq mutant in this work. Apart from this, virtually all proteins present in both the WT and mutant secretomes are upregulated in the Δ*hfq* condition. A closer look at the 30 proteins absent from the Δ*hfq* secretome also shows that they are all low-abundance proteins (median rank = 636). Taken together, this work supports the conclusion that the T4P system is not a main route for two-step secretion in cyanobacteria. Ultimately, given that the most abundant OM proteins are only permeable to inorganic ions[10], the main protein secretion channels across the OM remain unknown.

Protein secretion can also occur via "non-classical secretion". Proteins secreted by these mechanisms lack recognizable signal peptides or secretion motifs and include, for example, cytoplasmic proteins which can exert a second, "moonlighting", function outside the cell[63] and proteins secreted via extracellular vesicles[25,86]. Although non-classical secretion remains poorly characterized in cyanobacteria, a recent study has suggested that extracellular vesicles can be responsible for up to 10% of total protein secretion[25]. Taken together, these results suggest that many open questions remain regarding protein secretion in cyanobacteria.

In summary, our study provides a robust workflow that enables deep exploration of microbial exoproteomes. Application of this approach highlights cyanobacteria as unexpectedly proficient protein secreters and suggests the cyanobacterial secretome provides support to their successful photoautotrophic lifestyle. More broadly, our approach should be applicable to a wide range of bacteria and algae and could be readily adapted to field studies, thus opening new avenues in microbial exoproteomics.

## Data availability

The raw MS data and associated tables (i.e., peptide lists, unfiltered protein groups) have been deposited to the ProteomeXchange Consortium[87] via the PRIDE partner repository[88] and are publicly available with the identifier PXD047856. Fasta file input and tabular output from the BLAST reciprocal best hits analysis are available on Zenodo[89]. Source data for the figures in the manuscript are available in Supplementary Data files 1-15. All other data are available from the corresponding author on reasonable request.

## Code availability

Python code to reproduce the analysis and figures is publicly available via Zenodo[90].

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

## Acknowledgements

The authors thank Elke Dittmann for helpful discussions regarding *Nostoc* cultivation. We also thank Tanveer Batth for helpful discussions regarding method development and Felix Schneidmadel for mass spectrometry assistance. The authors acknowledge financial support from the Humboldt Foundation (D.A.R.), the Deutsche Forschungsgemeinschaft (DFG, German Research Foundation), SFB 1127 ChemBioSys, project number 239748522 (D.A.R., G.P., J.A.Z.Z.) and the Free state of Thuringia and the European Union via the "Innovationszentrum für Thüringer Medizintechnik-Lösungen" (ThIMEDOP; #2018 IZN 002) (F.M.).

## Author contributions

D.A.R.: Conceptualization, Methodology, Investigation, Formal analysis, Data curation, Visualization, Writing—Original draft, Writing—Review & editing, Resources, Funding acquisition. D.O.: Methodology, Investigation, Formal analysis, Data curation, Visualization, Writing—Original draft, Writing—Review & editing. G.P.: Writing—Review & editing, Resources, Funding acquisition. F.M.: Conceptualization, Methodology, Formal analysis, Writing—Review & editing, Resources, Funding acquisition. J.A.Z.Z.: Conceptualization, Methodology, Formal analysis, Visualization, Writing—Original draft, Writing—Review & editing, Resources, Funding acquisition. All authors have read and approved the final version of the manuscript. D.A.R. and D.O. contributed equally.

## Funding

## Competing interests

The authors declare no competing interests.
