## [Peer Review File · Communications Biology]

Reviewers' comments:

Reviewer #1 (Remarks to the Author):

Review of Russo et al., (2024)

This manuscript describes the use of a paramagnetic bead approach to concentrate proteins in the culture supernatants of three cyanobacterial species for subsequent proteomics analysis and with the additional development of a pipeline adapted to a 96-well plate high throughput format.

Protein secretion in cyanobacteria is perhaps less well understood compared to other bacteria primarily since, other than type I secretion and type IV pilus apparatus and potentially the ability to produce extracellular vesicles, these organisms seem to lack the classical type II, III, etc secretion systems. Thus, to some extent we are a bit in 'muddy water' with respect to cyanobacterial protein secretion since at least to me it is unclear whether some proteins that are present in culture supernatants are truly secreted via a specific active process or merely the result of non-specific cell lysis events or just the inherent 'leakiness' of these cells.

Thus, when reviewing this manuscript, I was particularly keen for the authors to demonstrate with this new methodology that it could indeed better differentiate proteins that are truly secreted versus those appearing as a result of leaky/cell lysis processes. In other words, are the authors trying to establish a method that is better at collecting information for more proteins present in the supernatant or the quality of the data produced - i.e. truly secreted proteins versus proteins that are present as a function of the methodology/cell lysis/leakiness events occurring? I don't really get an idea of the quality based on the data presented. Thus (line 341), it is mentioned that 885, 261, and 668 proteins were detected in culture supernatants of *Synechocystis*, *Synechococcus* and *Nostoc* respectively. Of these proteins 156, 117 and 81 were 'secreted' proteins based on bioinformatics predictions. (Just to say I disagree with the authors definition of secretome (line 366). Proteins are 'exported' to the periplasm and outer membrane and only 'secreted' proteins reach the extracellular media). As a percentage this showed that the new methodology could detect 62%, 46% and 14% of all potentially 'secreted' proteins in the respective organism's proteomes. So, what I really need to see with the new methodology is how such percentages compare to the previous literature and TCA precipitation and ultrafiltration approaches. Such data needs to be up front in the manuscript to give the reader a critical evaluation of the new bead approach versus previous technologies. Also, as mentioned above the number of 'secreted' proteins detected is very high. So, as well as better specifying the proportion of truly secreted proteins detected I think the manuscript also needs to specify in the text what is the % of ribosomal proteins or phycobiliprotein related proteins in the secretome to better assess cell lysis issues as opposed to true secretion (ideally compared to the % of these proteins in the soluble fraction) and how this compares between methodologies used.

The manuscript would also benefit markedly by disentangling the terminology used. Thus, the authors name the new methodology EXCRETE – for enhanced exoproteome characterisation. If the authors are indeed actually not trying to produce an improved technology to assess truly extracellular proteins i.e. those going OUTSIDE the cell, but rather aiming to assess exported

proteins generally (i.e. those crossing the cytoplasmic membrane to the periplasm and outer membrane (i.e. exported proteins) in addition to beyond to the extracellular environment (i.e. secreted proteins) then use of the term secretome/secreted should be removed and conclusions like those made in lines 30-33 qualified. For the latter the authors show no convincing evidence that the proteins they identify in the supernatant are truly secreted as opposed to leaking from the cell since there are a number of known cytoplasmic proteins in their extracellular datasets. The one caveat I would say on this point is the potential presence of extracellular vesicles (which e.g. in the cyanobacterium *Prochlorococcus* at least also possess known cytoplasmic proteins) but this would need a new study a) to show the organisms analysed indeed produced vesicles and b) what proteins these vesicles contain. That said, if the authors clarify their terminology and be consistent throughout the manuscript then the reader will be much more clear with the studies aims, and more able to judge the conclusions made.

Other comments:

Line 46: 'rich in salt, lipids and polysaccharides' is a strange way of stating the type of environment – please change to marine/hypersaline etc

Line 69: Please rephrase:widely used. However, these approaches...

Line 98: Please add a line describing how the CellDEG high density system works to produce aggregates

Line 234: Please rephrase: used solid phase. However, a variety...

Line 256: I assume this is a seawater concentration of salt - so rather than state salt rich (which to me implies hypersaline like conditions) I would just say seawater samples.

Line 269: Please mention here that 4.1% of proteins were only detected with ultrafiltration and not by EXCRETE.

Line 270: With respect to Fig 1E please state which dataset in the excel file contains the information describing these extracellular proteins.

Line 270: Where is the data on these low abundance proteins? Please specify or add.

Line 312: Where is the data on the unique proteins in each workflow that were of low abundance, and does this include abundance (spectral count info)? Please specify or add.

Line 341. Briefly explain the filtering and imputation that you performed here.

Line 343: mentions 'compared to previous studies'. Please cite the references related to these previous studies here.

Line 356: The coefficient of variation is shown in panel d not panel c. So, the figures need swapping.

Also, for panel c as presented what do you mean by relative protein identifications and relative sample number?

Line 360 I would advise to replace secretome with exoproteome and delete all use of the word secreted in this section and elsewhere in the manuscript. Just to say that *Synechocystis* does truly secrete a haemolysin (sll1951) (Sakiyama et al., 2006) but I couldn't see this in your exoproteome dataset – any thoughts?

Line 387: See my comments at the top of the review but how do these %s compare to other studies? i.e. % of exported proteins? (e.g. see Christie-Oleza et al., 2015 *Env Micro*).

Line 395: encodes a much larger number (i.e. delete for)

Line 398: for the *Nostoc* proteome how can it be excluded that there is not some technical problem with detecting proteins in the proteome of this organism e.g. excessive production of polysaccharides (so how does the proteome compare to the transcriptome - so is it known that a similar low fraction of genes are expressed at any one time?)

Lines 425-429: other cyanobacteria also produce such large proteins (see *Microbiology and Molecular Biology Reviews* 2009 73: 249-299 Giant proteins section) and this should be cited.

Line 434: Remarkably needs deleting here since I think this is over-stating this data

Lines 436-439: From Fig 4d the three highest ranking COG categories are hypothetical proteins, cell wall biogenesis and inorganic ion transport and metabolism – not as stated.

Line 440: Is this correct that signal transduction mechanisms show the largest deviation? To me the data shows that the % cell envelope biogenesis and hypothetical proteins are more variable between strains - the magnitude of variation is what's important here.

Lines 442-444: The COG categories may be conserved but what about conservation of protein orthologs in each category? Does each category comprise completely different proteins?

Line 464: The number considered truly secreted as a proportion of those detected in these two conditions is very different - any ideas why - is this a cell lysis issue?

Line 477-8: Fig 5C. Actually this data is in Fig 5e - and it looks like the inorganic ion and energy categories contribute the most to this difference. Arguably inorganic ion transport largely relates to proteins in the membrane rather than those secreted. Any thoughts? Also, I noticed 50S ribosomal protein L12 is highly differentially expressed in this condition and considered secreted - what is the reasoning behind the 'secretion' location?

Line 505: Fig 5e. This is the wrong figure in this panel and needs swapping with Fig 5c.

Lines 553-554: the statement 'a remarkable increase of up to eight times more' really needs to be deleted. This seems to be the number of proteins detected, many of which I would suggest are not truly secreted at all. Cell lysis cannot be considered a specific protein secretion mechanism. If it was being considered as so why would bacteria invest in specific secretion apparatus like type II, III etc systems?

Line 577: ...one-step T1SS. However, putative....

Line 623: Refs 4 and 14 require species names italicising

Reviewer #2 (Remarks to the Author):

The authors present a simultaneous protein aggregation and reduction/alkylation method, EXCRETE, for the purification of exoprotein from Cyanobacterial strains, in order to identify secreted proteins by mass spectrometry. They demonstrate that, compared to an ultrafiltration-based methodology, substantially more peptides and proteins are identified, with few proteins being 'missed' by the new methodology. EXCRETE is both rapid and functions on small quantities of starting material. Further, the authors demonstrate that EXCRETE can be modified to a higher throughput 96 well based format that has good agreement with the original microtube based method, with the potential for automation of the process, allowing for rapid screening of exoproteomic changes under different conditions. The authors then use EXCRETE to examine the secretome of three cyanobacterial species from different environments (marine, freshwater, and terrestrial), demonstrating the methods functionality and reproducibility across different conditions. Finally, EXCRETE is used to examine the secretome response of *Synechocystis* to three different conditions, showing a conserved core of secreted proteins that is supplemented by additional proteins under conditions representing a cyanobacterial bloom.

Originality and significance – To my knowledge this is original work. The EXCRETE methodology seems applicable to a range of questions involving exoproteins – I would certainly consider using it myself.

Data and methodology – The experimental strategy followed by the authors is valid. I have minor issues with data presentation, detailed below. Methodology is in sufficient detail for easy replication.

Statistics/uncertainty – Some figure legends require additional detail – see below – but statistical tests/reporting is appropriate and generally detailed enough

Conclusions – Ultimately, assigning proteins as exoproteins is based on predictive algorithms. Exoproteomics cannot definitively prove these predictions, as many intracellular proteins are also detected (Fig. 4a, b, and c) – though the most abundant proteins tend to be secreted. Nonetheless, the increase in the detection rate of predicted proteins is noteworthy, and the technique developed

within this work seems valuable.

Minor comments

Figure 3 legend - rework order of sections to reflect the order of the panels and main text. Consider reworking section e (referring to 3c) for clarity, e.g. percentage of protein groups present in at least 1, 2, etc replicates. 3d – Define the dashed and dotted lines in the legend. Coefficient of variation is presented as a percentage in 3d, but as a numerical value in 2b – is there a reason for the difference? Clarify what CV is based on – I assume Log₂ intensity values? 3e – clarify what the normalized rank is – I assume it is based on mean Log₂ intensity values? 3d – the colours used for Nostoc/Synechocystis aren't consistent with the other panels of the figure.

For the figures comparing the three Cyanobacterial strains, the choice of colours does not allow for easy discrimination between Synechocystis and Nostoc. Most panels consider the strains separately, so this is not an issue there, but Figure 3c, and to a greater extent 4d require more effort than necessary to interpret.

Figure 5 – Letters denoting panels c, d, and e appear in larger font than in other figures. 5d – Define the significance/ fold-change thresholds in the legend i.e. $p=0.05$ for significance etc.

Figure S1c Define the dashed and dotted lines in the legend

Figure S3 More clearly define the whiskers (If they are minimum/maximum values how do you have outliers?)

Figure S6 I'd like to see values on the axes

Line 360 – this is true at the level of broad functional categories, however it does not necessarily mean that secreted proteins within those categories are conserved between strains, so in the absence of evidence demonstrating that, could this be clarified? Also, the most abundant category is proteins of unknown function, which is likely to be a broad category! Within the most abundant COG categories I would expect conserved protein expression between species in COG IDs M (cell envelope biogenesis) and O (PTMs, protein turnover, chaperones), but more distinct expression profiles in COG IDs P (inorganic ion transport and metabolism) and S (function unknown). I'd be interested in further exploration of this dataset, though I'm not sure I'd consider it necessary for the manuscript.

Reviewer #3 (Remarks to the Author):

The manuscript by Russo et. al., describes a novel workflow for proteomic analysis of secreted proteins, The technique leverages solid phase enhanced protein aggregation protocols. The study employs this protocol and compares its efficacy to more traditional approaches, such as ultrafiltration for protein concentration using three model cyanobacteria in their analysis. The

results provide strong evidence supporting the efficacy of this approach. Generally, the paper is well written, and the results support the conclusions. There are however several points where the authors could elaborate or clarify their results to better connect them to the biology of study organisms. Additionally, there are some points where more detailed data presentation would be useful. Please see specific comments below for details.

Line 361 – it might be worth adding outer membrane vesicles to this list

Line 394-401 – *N. punctiforme* exhibits several developmental cell types. It would be important to know what cell types were present during this analysis. Does the medium contain fixed nitrogen, and if so had the fixed nitrogen been depleted (did the authors check to see if heterocysts were forming in the culture)? Did the culture contain motile hormogonia? Akinetes? Without this information it is difficult to assess the exoproteome results. If the authors did not do this analysis, they should clarify that it is a limitation and that the absence of various cell types might contribute to the low levels of secreted proteins.

Line 410-413 – While this is a possible explanation for the lack of PilA, another is that there were no hormogonia under the culture conditions. PilA is only expressed in hormogonia and this could account for its absence. Again, it is critical to know the state of the culture to explain this.

Line 458-460 – It would be useful for the authors to clarify the role of hfq here. Hfq directly interacts with PilB and is essential for type IV pilus extension. The phenotypes of hfq mutants in cyanobacteria (loss of motility, competence, aggregation) are due to the loss of T4P function and has important implications for the exoproteome analysis.

Line 481-482 – It would be useful to include a list of the proteins that are absent from the hfq mutant exoproteome, I could not seem to find one. Although I would imagine it can be extracted through the available supplemental data, it would be nice to provide this information more directly to the reader. Given that PilA would no longer be exported across the outer membrane in the hfq mutant one would expect PilA to be missing or dramatically reduced in the hfq exoproteome. Was this the case?

571-573 – Enzymes involved in cell envelope assembly and maintenance would presumably be ubiquitous to all bacteria/living organisms in general. It is not clear why the authors single out cyanobacteria here.

Line 582-583 – Some have speculated that the cyanobacterial T4P system also functions as a T2SS? Would it be worth discussing this in the context of this study? In particular, how does this relate to the data on the hfq mutant, given that mutation of hfq inactivates the T4P system. Again, it would be worth discussing the proteins that are differentially secreted in the hfq mutant vs wild type with this context in mind.

We would like to thank the reviewers for the careful reading of our work and their constructive comments, which helped us to strengthen our manuscript substantially. Please find our detailed answers to each point below marked in blue.

Reviewer #1 (Remarks to the Author):

Review of Russo et al., (2024)

This manuscript describes the use of a paramagnetic bead approach to concentrate proteins in the culture supernatants of three cyanobacterial species for subsequent proteomics analysis and with the additional development of a pipeline adapted to a 96-well plate high throughput format.

Protein secretion in cyanobacteria is perhaps less well understood compared to other bacteria primarily since, other than type I secretion and type IV pilus apparatus and potentially the ability to produce extracellular vesicles, these organisms seem to lack the classical type II, III, etc secretion systems. Thus, to some extent we are a bit in 'muddy water' with respect to cyanobacterial protein secretion since at least to me it is unclear whether some proteins that are present in culture supernatants are truly secreted via a specific active process or merely the result of non-specific cell lysis events or just the inherent 'leakiness' of these cells.

Thus, when reviewing this manuscript, I was particularly keen for the authors to demonstrate with this new methodology that it could indeed better differentiate proteins that are truly secreted versus those appearing as a result of leaky/cell lysis processes. In other words, are the authors trying to establish a method that is better at collecting information for more proteins present in the supernatant or the quality of the data produced - i.e. truly secreted proteins versus proteins that are present as a function of the methodology/cell lysis/leakiness events occurring?

We fully agree with the reviewer that there is an unmet need to study protein secretion in cyanobacteria in more detail. To address this, we established a novel mass spectrometry-based method that identifies and quantifies proteins in the supernatant at unprecedented depth. The reviewer is correct that this encompasses both 'truly' secreted proteins as well as those resulting from unspecific processes, which is an inherent challenge in 'exoproteomics' or 'secretomics'. This is because the origin of a protein detected by mass spectrometry is, in general, unknown. Except in a handful of circumstances (e.g., the presence of a specific motif), there is not a method that can be selective towards "truly secreted" proteins. Therefore, our study strives to increase the ability to map what is outside the cell.

I don't really get an idea of the quality based on the data presented. Thus (line 341), it is mentioned that 885, 261, and 668 proteins were detected in culture supernatants of *Synechocystis*, *Synechococcus* and *Nostoc* respectively. Of these proteins 156, 117 and 81 were 'secreted' proteins based on bioinformatics predictions. (Just to say I disagree with the authors definition of secretome (line 366). Proteins are 'exported' to the periplasm and outer membrane and only 'secreted' proteins reach the extracellular media).

The manuscript would also benefit markedly by disentangling the terminology used. Thus, the authors name the new methodology EXCRETE – for enhanced exoproteome characterisation. If the authors are indeed actually not trying to produce an improved technology to assess truly extracellular proteins i.e. those going OUTSIDE the cell, but rather aiming to assess exported proteins generally (i.e. those crossing the cytoplasmic membrane to the periplasm and outer membrane (i.e. exported proteins) in addition to beyond to the extracellular environment (i.e. secreted proteins) then use of the term secretome/secreted should be removed and conclusions like those made in lines 30-33 qualified. For the latter the authors show no convincing evidence that the proteins they identify in the supernatant are truly secreted as opposed to leaking from the cell since there are a number of known cytoplasmic proteins in their extracellular datasets. The one caveat I would say on this point is the potential presence of extracellular vesicles (which e.g. in the cyanobacterium *Prochlorococcus* at least also possess known cytoplasmic proteins) but this would need a new study a) to show the organisms analysed indeed produced vesicles and b) what proteins these vesicles contain. That said, if the authors clarify their terminology and be consistent throughout the manuscript then the reader will be much more clear with the studies aims, and more able to judge the conclusions made.

Both reviewer comments concern the terminology used, therefore we answer them together. The points raised are valid and in the following we explain our rationale behind the chosen terminology and explain how we have attempted to conciliate all viewpoints.

We acknowledge that there is a longstanding debate within the community regarding secretion terminology. Our definition of "secretome" follows that of Tsirigotaki et al., 2017 (DOI: 10.1038/nrmicro.2016.161, Nature Reviews Microbiology).

“The portion of the bacterial exportome that is exported beyond the plasma membrane (that is, to the periplasm, outer membrane, extracellular milieu or a host cell; approximately 13% of the total proteome of Escherichia coli K12).”

We are aware that this definition does not perfectly align with the classic and more stringent definition proposed, for example, in Desvaux et al., 2009 (DOI: 10.1016/j.tim.2009.01.004, Trends in Microbiology).

“To make it even more explicit, we propose defining a secreted protein as a protein transported via a secretion system – therefore, the term ‘secreted protein’ does not refer exclusively to those proteins present in the extracellular milieu but includes all of those existing outside of the outermost lipid bilayer, as described above.”

The most likely intention of Tsirigotaki et al., 2017 was to update the terms to encompass:

- 1) Gram-positive bacteria where secretion occurs via Sec and Tat and
- 2) secretion via pathways that are not recognised as canonical “secretion systems” (e.g., exosortases that can surface display proteins with a PEP-CTERM motif) and
- 3) embrace the fact that the periplasmic and extracellular space show some fluidity and exchange. Many proteins exist both in the periplasm and the extracellular space, occasionally even with different functions (Giner-Lamia et al., 2016 DOI:10.3389/fmicb.2016.00878 Extracellular Proteins: Novel Key Components of Metal Resistance in Cyanobacteria?).

With the additional evidence provided below that our exoproteome is not primarily a result of cell lysis, we have decided to stay closer to the definition of Tsirigotaki et al., 2017. However, we have softened several statements to include terms such as “predicted secretome” and “predicted as secreted” and avoided stating we have identified secreted proteins or the secretome. We also added a sentence to make the readers aware of these semantic questions and clearly define the terminology used in our manuscript.

“Regarding location, historically, a secreted protein was defined as one exclusively transported via a secretion system (Desvaux, Hébraud, Talon & Henderson 2009). However, this fails to encompass the known complexity of the Gram-negative cell envelope where a protein can be active in both the periplasm and the extracellular space, and can be secreted via non-canonical secretion systems. Therefore, we have adopted a less stringent definition of secretome encompassing proteins predicted to be translocated to the periplasm, outer membrane or extracellular milieu (Tsirigotaki, De Geyter, Šoštarić, Economou & Karamanou 2017).”

We hope that these revisions clarify the aims and scope of our study.

As a percentage this showed that the new methodology could detect 62%, 46% and 14% of all potentially ‘secreted’ proteins in the respective organism’s proteomes. So, what I really need to see with the new methodology is how such percentages compare to the previous literature and TCA precipitation and ultrafiltration approaches. Such data needs to be up front in the manuscript to give the reader a critical evaluation of the new bead approach versus previous technologies.

Following the reviewer’s suggestion, we added this new analysis to the manuscript at the top of the section predicting secreted proteins. We also added the corresponding results from the literature mentioned below to Supplementary figure 2.

“Following the criteria defined above, we detected 156, 117 and 81 proteins predicted to be secreted in Synechocystis, Synechococcus, and Nostoc (corresponding to 62%, 46% and 14% of all potentially secreted proteins in the respective proteomes) (Supplementary Data 4, 8, 9). Applying the same criteria to published datasets showed that, in Synechocystis, only up to 8% of potentially secreted proteins have been previously identified using ultrafiltration-based sample preparation (Supplementary Fig. 2c) (Oliveira et al. 2016). In Nostoc, we found a better overlap with a study using precipitation-based sample preparation (13% of potentially secreted proteins (76 proteins) (Warshan et al. 2017) vs 14% (81 proteins) in this study (Supplementary Fig. 2d)).”

Also, as mentioned above the number of ‘secreted’ proteins detected is very high. So, as well as better specifying the proportion of truly secreted proteins detected I think the manuscript also needs to specify in the text what is the % of ribosomal proteins or phycobiliprotein related proteins in the secretome to better assess cell lysis issues as opposed to true secretion (ideally compared to the % of these proteins in the soluble fraction) and how this compares between methodologies used.

Thank you for suggesting this additional analysis of intracellular ‘marker’ proteins. Unfortunately we have not found a cyanobacterial study with the species we used here that published an exoproteome with its

corresponding endoproteome. In our study we have the matching endo/exoproteome for *Synechocystis* and *Nostoc* analysed with the EXCRETE method and that allowed us to do the analysis the reviewer suggested. The table below shows a tally of proteins considered as intracellular markers. Percentages correspond to the intensity of the group of proteins relative to the total intensity of the endo or exo fraction.

We have not found a study on the exoproteome of the species analysed here that has reported protein intensities. However, as the reviewer suggests in a comment below, this analysis can be done for Christie-Oleza et al., 2015 Env Micro and Kaur et al., 2018 Env Micro. Here the authors also report quantification data (normalized spectral counts) and relative abundance (individual protein counts as a percentage of total counts). We also included this data in the same supplementary table and in short form in the main manuscript.

Species	Photosystem subunits	Phycobilins	Ribosomes	RuBisCO	Reference
Endoproteome					
Synechocystis sp. PCC 6803	12.1%	17.2%	4.7%	3.5%	This work
Nostoc punctiforme PCC 73102	6.2%	6.4%	15.9%	3.1%	This work
Exoproteome					
Synechocystis sp. PCC 6803	1.5%	5.9%	1.1%	0.1%	This work
Synechococcus sp. PCC 11901	0.3%	1.1%	0.0%	0.0%	This work
Nostoc punctiforme PCC 73102	0.6%	2.0%	6.0%	1.2%	This work
Synechococcus sp. WH 5701	3.0%	9.3%	1.4%	0.1%	Christie-Oleza et al., 2015
Synechococcus sp. WH 7803	2.3%	26.1%	0.6%	0.1%	Christie-Oleza et al., 2015
Synechococcus sp. WH 7805	0.0%	12.9%	0.0%	0.0%	Christie-Oleza et al., 2015
Synechococcus sp. WH 8102	2.8%	37.3%	1.1%	0.0%	Christie-Oleza et al., 2015

“In addition, to investigate to which extent proteins identified outside the cell stem from unspecific processes such as cell lysis, we evaluated the relative abundance of intracellular marker proteins (photosystem subunits, phycobilins, ribosomes, and RuBisCO) in the detected exoproteomes of the three cyanobacterial species. These proteins accounted for 9%, 2% and 10% of the total protein intensity in *Synechocystis*, *Synechococcus*, and *Nostoc*, respectively (Supplementary Table 1). In comparison, the intensity of these proteins accounted for 38% and 32% of the *Synechocystis* and *Nostoc* endoproteomes, respectively (Supplementary Data 10, 11). These numbers compare favorably to previous studies of marine *Synechococcus* species where these intracellular markers accounted for 14 – 50% of the respective exoproteomes (Christie-Oleza, Armengaud, Guerin & Scanlan 2015; Kaur, Hernandez-Fernaud, Aguilo-Ferretjans, Wellington & Christie-Oleza 2018).”

Other comments:

Line 46: ‘rich in salt, lipids and polysaccharides’ is a strange way of stating the type of environment – please change to marine/hypersaline etc

“...and can be found in freshwater, marine, and hypersaline environments, as well as in biofilms and microbial mats”.

Line 69: Please rephrase:widely used. However, these approaches...

Done.

Line 98: Please add a line describing how the CellDEG high density system works to produce aggregates

A sentence was added “The addition of CO₂, together with the slow shaking, promotes the aggregating phenotype.”

Line 234: Please rephrase: used solid phase. However, a variety...

Done.

Line 256: I assume this is a seawater concentration of salt - so rather than state salt rich (which to me implies hypersaline like conditions) I would just say seawater samples.

Changed to “seawater samples”.

Line 269: Please mention here that 4.1% of proteins were only detected with ultrafiltration and not by EXCRETE.
Line 270: Where is the data on these low abundance proteins? Please specify or add.

Changed to “Finally, regarding protein identifications, we observed an overlap of 47% of proteins with 49% only identified with EXCRETE and 4.1% exclusive to the ultrafiltration workflow (Fig. 1e, Supplementary Data 1, Supplementary Data 2).”

We have added an additional table (Supplementary Data 3) with the unique proteins and their abundances. Additionally, a sentence was added stating that the median rank of the unique proteins is 348.

The additional proteins identified included several low abundance proteins (median rank of unique proteins = 348), thus demonstrating the robustness of our method across a wide dynamic range (Supplementary Fig. 1b).”

Line 270: With respect to Fig 1E please state which dataset in the excel file contains the information describing these extracellular proteins.

A reference to the supplementary datasets 1 and 2 was added.

Line 312: Where is the data on the unique proteins in each workflow that were of low abundance, and does this include abundance (spectral count info)? Please specify or add.

We have added separate tables with the unique proteins for each workflow (Supplementary Data 6 and 7). Additionally, a sentence was added stating that the median rank of the unique proteins in the microtube workflow is 201 and in the microplate workflow is 215.

“The unique proteins identified in each workflow were generally of lower abundance (median rank of unique proteins = 215 in microtubes and 201 in microplates) (Fig. 2f, Supplementary Data 6 and 7).”

Line 341. Briefly explain the filtering and imputation that you performed here.

The following sentence was added “The raw data was then filtered (only protein groups present in at least 70% of the replicates with a minimum of three replicates were retained) and imputed using the k-nearest neighbors algorithm.”. This information is also available in the “Statistical Analysis” section of the Methods.

Line 343: mentions ‘compared to previous studies’. Please cite the references related to these previous studies here.

We added the references from Supplementary Figure 2 to the main text.

Line 356: The coefficient of variation is shown in panel d not panel c. So, the figures need swapping. Also, for panel c as presented what do you mean by relative protein identifications and relative sample number?

The panels are correctly ordered as described in the text but the figure legend was, indeed, incorrect. This has been corrected.

Line 360 I would advise to replace secretome with exoproteome and delete all use of the word secreted in this section and elsewhere in the manuscript. Just to say that *Synechocystis* does truly secrete a haemolysin (sll1951) (Sakiyama et al., 2006) but I couldn’t see this in your exoproteome dataset – any thoughts?

Regarding the first part of the comment, we have addressed this at length above. In sum, we have softened several statements to include terms such as “predicted secretome” and “predicted as secreted” and avoided stating we have identified secreted proteins or the secretome. We also added a sentence to clearly define the terminology used in our manuscript.

Regarding sll1951, in this study we use a motile strain of *Synechocystis*.

From the abstract of Sakiyama et al., 2006: “In some other cyanobacteria, RTX proteins are reported to be necessary for cell motility. However, the GT was immotile. Moreover, the motile wild-type strain did not express any HLP, suggesting that HLP is one of the factors involved in the elimination of motility in the GT.”

In a later study (Trautner and Vermaas, 2016 DOI: 10.1128/JB.00615-13): “Sll1951 is the surface layer (S-layer) protein of the cyanobacterium *Synechocystis* sp. strain PCC 6803. This large, hemolysin-like protein was found in the supernatant of a strain that was deficient in S-layer attachment.”

We avoided discussion of the strain differences because 1) it is often unclear what strains are used in each particular study and 2) it is well-known (e.g., Schuergers and Wilde, 2015 DOI: 10.3390/life5010700) that the non-motile lineage of *Synechocystis* has multiple secretion deficiencies therefore should be avoided for secretion studies.

Line 387: See my comments at the top of the review but how do these %s compare to other studies? i.e. % of exported proteins? (e.g. see Christie-Oleza et al., 2015 Env Micro).

See details in comment above.

Line 395: encodes a much larger number (i.e. delete for)

Done.

Line 398: for the *Nostoc* proteome how can it be excluded that there is not some technical problem with detecting proteins in the proteome of this organism e.g. excessive production of polysaccharides (so how does the proteome compare to the transcriptome - so is it known that a similar low fraction of genes are expressed at any one time?)

While we cannot definitively exclude technical issues, the excessive production of polysaccharides is unlikely to hinder protein detection. In the following section we extract a large number of exoproteins from an aggregated *Synechocystis* where polysaccharide accumulation makes the supernatant highly viscous.

Regarding comparisons with the transcriptome we are cautious. It is well-known that more often than not the transcriptome and proteome do not correlate well. For example, Campbell et al, 2008 (DOI: 10.1128/JB.00990-08) report "This analysis yielded expression signals from 2,935 genes, nearly twice the number of genes for the soluble NH₄⁺-grown proteome (1,572 genes)".

This notwithstanding, the numbers reported here are comparable to ours and this reference has been added to the manuscript. In addition, our numbers also compare well with other more recent proteomics studies such as Warshan et al., 2017 and Álvarez et al., 2022.

Lines 425-429: other cyanobacteria also produce such large proteins (see Microbiology and Molecular Biology Reviews 2009 73: 249-299 Giant proteins section) and this should be cited.

The following sentence was added "Large proteins, reaching up to 2.72 MDa, have also previously been identified in multiple marine and freshwater cyanobacteria (Scanlan et al. 2009; Cheregi, Miranda, Gröbner & Funk 2015)."

Line 434: Remarkably needs deleting here since I think this is over-stating this data

"Remarkably" has been deleted.

Lines 436-439: From Fig 4d the three highest ranking COG categories are hypothetical proteins, cell wall biogenesis and inorganic ion transport and metabolism – not as stated.

These statements were qualified in the previous sentence "Amongst the proteins with known COG functions..." therefore the category "Hypothetical proteins" was not included.

Line 440: Is this correct that signal transduction mechanisms show the largest deviation? To me the data shows that the % cell envelope biogenesis and hypothetical proteins are more variable between strains - the magnitude of variation is what's important here.

As explained above, this analysis refers only to proteins with known COG functions. Amongst these, cell envelope biogenesis varies from 9.44% to 19.1% while signal transduction varies from 2.25% to 6.67%. It can be debated whether the absolute or relative variation is more important here therefore we included statements for both.

"Amongst all the categories, signal transduction mechanisms showed the largest relative deviation, with this category being represented almost four times higher in *Synechocystis* than in *Synechococcus* and *Nostoc*. In absolute terms, the largest deviation was seen for cell wall/membrane/envelope biogenesis with a difference of nearly 10% between species."

Lines 442-444: The COG categories may be conserved but what about conservation of protein orthologs in each

category? Does each category comprise completely different proteins?

This is an excellent point which was also raised by reviewer 2. Therefore, we undertook a reciprocal best hits BLAST analysis to investigate the conservation of protein orthologs in each category. We have added further detail regarding this comment in the results and the discussion section.

In the results:

“Given that the functional profile was conserved across all three species, we proceeded to investigate the degree of conservation between the proteins predicted as secreted within each of the top COG categories. To this end, we used BLAST to map the reciprocal best hits between each category from each species and the same category in the other two species. The results showed that the posttranslational modification, protein turnover, chaperones, and cell wall/membrane/envelope biogenesis were the most conserved categories with an average of 45% and 55%, respectively, of proteins having orthologs in at least one other species. The categories unknown function and inorganic ion transport and metabolism were less conserved with only 28% and 23%, respectively, of proteins having orthologs in at least one other species (Supplementary Table 3). Altogether, these results suggest that cell envelope maintenance and nutrient acquisition are core functions of the predicted secretomes albeit with different levels of ortholog conservation across cyanobacteria from different habitats.”

In the discussion:

“A closer look at the individual proteins across species showed that multiple protein orthologs present in all three species were identified in the categories of protein turnover and cell envelope management. In the former category, the widely conserved Deg/HtrA endopeptidases HhoA and HhoB were identified. In the latter category, porins (e.g., with SLH/OprB-domains) and enzymes such as N-acetylmuramoyl-L-alanine amidases, lytic transglycosylases, and peptidyl-prolyl cis-trans isomerases were identified in all species. A recent study in *Anabaena* sp. PCC 7120 also identified several extracellular amidases with potential roles in peptidoglycan recycling and the formation of nanopores (Sarasa-Buisan et al. 2024). This suggests that enzymes involved in cell envelope assembly and maintenance are ubiquitous and are a promising direction of future studies in cyanobacteria. The categories of nutrient acquisition and unknown function contained less orthologs shared in all three species. In the former category, this is a result of different nutrient preferences. *Synechococcus* showed a specialization in Fe metabolism while the *Synechocystis* and *Nostoc* exoproteomes contained more proteins related to P, S and Zn metabolism. In the latter category, we could not determine the function of the shared orthologs but the beta lactamase fold, pentapeptide repeats and tetratricopeptide repeat motifs were present in all three species.”

Line 464: The number considered truly secreted as a proportion of those detected in these two conditions is very different - any ideas why - is this a cell lysis issue?

Given we don't have a definitive explanation we did not speculate as to why this might be the case. But our hypothesis is that the extreme viscosity of the medium leads to lower levels of protein turnover so we see a higher abundance of all proteins. This, naturally, leads to more identifications of lower abundance proteins. An analysis of the top 100 proteins shows that in standard conditions 47 are “secreted” and in bloom conditions 44 are “secreted” with a lot of overlap between both conditions.

Line 477-8: Fig5C. Actually this data is in Fig 5e - and it looks like the inorganic ion and energy categories contribute the most to this difference.

This is correct and has been included in the text.

“In agreement with our hypothesis, we observed a combined 26% increase in the inorganic ion transport, energy production, and amino acid transport categories...”

Arguably inorganic ion transport largely relates to proteins in the membrane rather than those secreted. Any thoughts?

Looking at the dataset, the proteins in the category of ion transport unique to the bloom condition are mostly annotated as periplasmic (see table below). These proteins relate to iron, nickel and manganese metabolism. These systems will definitely have an associated membrane protein but it is typical in shotgun proteomics experiments to mostly identify the periplasmic components. What is interesting here is that many of these periplasmic components are also proposed to have an extracellular role (reviewed in Giner-Lamia et al., 2016 DOI:10.3389/fmicb.2016.00878) but this is still poorly understood. We hope that our workflow will contribute to shed light on this in the future.

protein_name	locus_tag_kaz	desc_eggnog
iron(III) dicitrate transport system permease protein FecB	slr1319	Periplasmic binding protein
hypothetical protein	sll0382	PFAM Nickel transport complex, NikM subunit, transmembrane
ferrichrome-iron receptor	sll1406	TonB dependent receptor
iron(III) dicitrate-binding periplasmic protein	slr1492	Periplasmic binding protein
hypothetical protein	sll0237	ABC-type Fe3 transport system, periplasmic component
Mn transporter MntC	sll1598	Zinc-uptake complex component A periplasmic
ferric aerobactin receptor	sll1206	TonB dependent receptor
iron(III) dicitrate-binding periplasmic protein	slr1491	COGs COG0614 ABC-type Fe3 - hydroxamate transport system periplasmic component
molybdate-binding periplasmic protein	sll0738	Bacterial extracellular solute-binding protein
ferrichrome-iron receptor	slr1490	COGs COG4773 Outer membrane receptor for ferric coprogen and ferric-rhodotorulic acid

Also, I noticed 50S ribosomal protein L12 is highly differentially expressed in this condition and considered secreted - what is the reasoning behind the 'secretion' location?

After looking at our prediction databases we noticed that the hidden markov model that identifies signal peptides in PsortB has labelled this protein as "non-cytoplasmic with a signal peptide". However, given that this protein belongs to the ribosomes, and this annotation is in disagreement with UniprotKB, it is likely misannotated. As with any algorithm there will be false positives and false negatives and we refrained from manually curating every prediction to not introduce any human bias.

Line 505: Fig 5e. This is the wrong figure in this panel and needs swapping with Fig 5c.

This is correct. Panels c and e in figure 5 have been swapped.

Lines 553-554: the statement 'a remarkable increase of up to eight times more' really needs to be deleted. This seems to be the number of proteins detected, many of which I would suggest are not truly secreted at all. Cell lysis cannot be considered a specific protein secretion mechanism. If it was being considered as so why would bacteria invest in specific secretion apparatus like type II, III etc systems?

As we have shown above, this is also true for the fraction of all potentially secreted proteins in the *Synechocystis* proteome (8% vs 62%). Therefore, we have slightly modified the statement but kept the comparison.

"In comparison to previous studies, application of EXCRETE to three different cyanobacteria enabled the detection of a significantly higher number of proteins predicted to be secreted (Supplementary Fig. 2c, d)."

Line 577: ...one-step T1SS. However, putative....

Done.

Line 623: Refs 4 and 14 require species names italicising

Done.

Reviewer #2 (Remarks to the Author):

The authors present a simultaneous protein aggregation and reduction/alkylation method, EXCRETE, for the purification of exoprotein from Cyanobacterial strains, in order to identify secreted proteins by mass spectrometry.

They demonstrate that, compared to an ultrafiltration-based methodology, substantially more peptides and proteins are identified, with few proteins being 'missed' by the new methodology. EXCRETE is both rapid and functions on small quantities of starting material. Further, the authors demonstrate that EXCRETE can be modified to a higher throughput 96 well based format that has good agreement with the original microtube based method, with the potential for automation of the process, allowing for rapid screening of exoproteomic changes under different conditions. The authors then use EXCRETE to examine the secretome of three cyanobacterial species from different environments (marine, freshwater, and terrestrial), demonstrating the methods functionality and reproducibility across different conditions. Finally, EXCRETE is used to examine the secretome response of *Synechocystis* to three different conditions, showing a conserved core of secreted proteins that is supplemented by additional proteins under conditions representing a cyanobacterial bloom.

Originality and significance – To my knowledge this is original work. The EXCRETE methodology seems applicable to a range of questions involving exoproteins – I would certainly consider using it myself.

Data and methodology – The experimental strategy followed by the authors is valid. I have minor issues with data presentation, detailed below. Methodology is in sufficient detail for easy replication.

Statistics/uncertainty – Some figure legends require additional detail – see below – but statistical tests/reporting is appropriate and generally detailed enough

Conclusions – Ultimately, assigning proteins as exoproteins is based on predictive algorithms. Exoproteomics cannot definitively prove these predictions, as many intracellular proteins are also detected (Fig. 4a, b, and c) – though the most abundant proteins tend to be secreted. Nonetheless, the increase in the detection rate of predicted proteins is noteworthy, and the technique developed within this work seems valuable.

Minor comments

Figure 3 legend - rework order of sections to reflect the order of the panels and main text.

The panels are correctly ordered as described in the text but the figure legend was, indeed, incorrect. This has been changed.

Consider reworking section e (referring to 3c) for clarity, e.g. percentage of protein groups present in at least 1, 2, etc replicates.

The figure has been reworked to increase clarity and the figure legend now reads:

“c Percentage of protein groups present with an increasing number of replicates for each species.”

3d – Define the dashed and dotted lines in the legend.

Dashed line represents the median. Dotted lines represent the top and bottom quartiles. This has been added to the legend.

Coefficient of variation is presented as a percentage in 3d, but as a numerical value in 2b – is there a reason for the difference? Clarify what CV is based on – I assume Log₂ intensity values?

There is no reason for the difference in depiction of the CV. Figure 2b should also have the axis in percentage and that has now been updated. CVs are calculated on the non-transformed abundance values to show the true variation of the data. This has been clarified in the figure legend.

3e – clarify what the normalized rank is – I assume it is based on mean Log₂ intensity values?

The assumption is correct and this has been clarified.

3d – the colours used for *Nostoc*/*Synechocystis* aren't consistent with the other panels of the figure.

Thank you for noticing, this has been fixed.

For the figures comparing the three Cyanobacterial strains, the choice of colours does not allow for easy discrimination between *Synechocystis* and *Nostoc*. Most panels consider the strains separately, so this is not an issue there, but Figure 3c, and to a greater extent 4d require more effort than necessary to interpret.

The colors have been changed to improve the contrast between the blue and green.

Figure 5 – Letters denoting panels c, d, and e appear in larger font than in other figures.

This has been corrected.

5d – Define the significance/ fold-change thresholds in the legend i.e. $p=0.05$ for significance etc.

This has been changed to “The vertical and horizontal dashed lines represent the fold-change (plotted at -1 and 1) and significance (plotted at 0.05) thresholds, respectively.”

Figure S1c Define the dashed and dotted lines in the legend

Done.

Figure S3 More clearly define the whiskers (If they are minimum/maximum values how do you have outliers?)

This was a typo missed during manuscript preparation. The whiskers have been plotted according to the Tukey method and do not represent minimum and maximum values. This has been clarified in the figure legend.

Figure S6 I'd like to see values on the axes

Values have been added.

Line 360 – this is true at the level of broad functional categories, however it does not necessarily mean that secreted proteins within those categories are conserved between strains, so in the absence of evidence demonstrating that, could this be clarified? Also, the most abundant category is proteins of unknown function, which is likely to be a broad category! Within the most abundant COG categories I would expect conserved protein expression between species in COG IDs M (cell envelope biogenesis) and O (PTMs, protein turnover, chaperones), but more distinct expression profiles in COG IDs P (inorganic ion transport and metabolism) and S (function unknown). I'd be interested in further exploration of this dataset, though I'm not sure I'd consider it necessary for the manuscript.

This is an excellent point which was also raised by reviewer 1. We have added a reciprocal best hits BLAST analysis investigating the conservation of protein orthologs in each category. Further detail has been added in the results and the discussion section.

In the results:

“Given that the functional profile was conserved across all three species, we proceeded to investigate the degree of conservation between the proteins predicted as secreted within each of the top COG categories. To this end, we used BLAST to map the reciprocal best hits between each category from each species and the same category in the other two species. The results showed that the posttranslational modification, protein turnover, chaperones, and cell wall/membrane/envelope biogenesis were the most conserved categories with an average of 45% and 55%, respectively, of proteins having orthologs in at least one other species. The categories unknown function and inorganic ion transport and metabolism were less conserved with only 28% and 23%, respectively, of proteins having orthologs in at least one other species (Supplementary Table 3). Altogether, these results suggest that cell envelope maintenance and nutrient acquisition are core functions of the predicted secretomes albeit with different levels of ortholog conservation across cyanobacteria from different habitats.”

In the discussion:

“A closer look at the individual proteins across species showed that multiple protein orthologs present in all three species were identified in the categories of protein turnover and cell envelope management. In the former category, the widely conserved Deg/HtrA endopeptidases HhoA and HhoB were identified. In the latter category, porins (e.g., with SLH/OprB-domains) and enzymes such as N-acetylmuramoyl-L-alanine amidases, lytic transglycosylases, and peptidyl-prolyl cis-trans isomerases were identified in all species. A recent study in *Anabaena* sp. PCC 7120 also identified several extracellular amidases with potential roles in peptidoglycan recycling and the formation of nanopores (Sarasa-Buisan et al. 2024). This suggests that enzymes involved in cell envelope assembly and maintenance are ubiquitous and are a promising direction of future studies in cyanobacteria. The categories of nutrient acquisition and unknown function contained less orthologs shared in all three species. In the former category, this is a result of different nutrient preferences. *Synechococcus* showed a specialization in Fe metabolism while the *Synechocystis* and *Nostoc* exoproteomes contained more proteins related to P, S and Zn metabolism. In the latter category, we could not determine the function of the shared

orthologs but the beta lactamase fold, pentapeptide repeats and tetratricopeptide repeat motifs were present in all three species.”

Reviewer #3 (Remarks to the Author):

The manuscript by Russo et. al., describes a novel workflow for proteomic analysis of secreted proteins, The technique leverages solid phase enhanced protein aggregation protocols. The study employs this protocol and compares its efficacy to more traditional approaches, such as ultrafiltration for protein concentration using three model cyanobacteria in their analysis. The results provide strong evidence supporting the efficacy of this approach. Generally, the paper is well written, and the results support the conclusions. There are however several points where the authors could elaborate or clarify their results to better connect them to the biology of study organisms. Additionally, there are some points where more detailed data presentation would be useful. Please see specific comments below for details.

Line 361 – it might be worth adding outer membrane vesicles to this list

Agreed and added.

Line 394-401 – *N. punctiforme* exhibits several developmental cell types. It would be important to know what cell types were present during this analysis. Does the medium contain fixed nitrogen, and if so had the fixed nitrogen been depleted (did the authors check to see if heterocysts were forming in the culture)? Did the culture contain motile hormogonia? Akinetes? Without this information it is difficult to assess the exoproteome results. If the authors did not do this analysis, they should clarify that it is a limitation and that the absence of various cell types might contribute to the low levels of secreted proteins.

This is a great point. Based on our microscopic observations the cells are mostly in a vegetative state with some heterocysts starting to appear. We did not observe other developmental cell types. The growth medium used contains nitrogen (standard BG-11). However, it's likely the nitrogen became limiting during cultivation. As the reviewer mentions, this lack of complexity might explain the low levels of secreted proteins so this has been added to the manuscript. We also added representative microscopy pictures of *Nostoc* cultures to the supplementary material (Supp. Fig. 3)

“This is in line with previous work showing that multicellular symbiotic cyanobacteria, such as *Nostoc*, exhibit high levels of proteomic plasticity and only express a small fraction of their proteome at any given point in time (Campbell, Summers, Christman, Martin & Meeks 2007; Warshan et al. 2017; Vries & Vries 2022; Baunach, Guljamow, Miguel-Gordo & Dittmann 2023). Accordingly, our microscopic observations showed that the *Nostoc* culture was mostly in a vegetative state with only a few heterocysts (Supplementary Fig. 3). Therefore, the absence of different cell types might also explain the low number of protein identifications.”

Line 410-413 – While this is a possible explanation for the lack of PilA, another is that there were no hormogonia under the culture conditions. PilA is only expressed in hormogonia and this could account for its absence. Again, it is critical to know the state of the culture to explain this.

We appreciate this insight. Based on the pictures collected we indeed could not identify any hormogonia. Thus, this is the most likely reason for the lack of PilA1 and the manuscript has been updated accordingly.

“Next, PilA1, the major component of the retractable fiber of the T4P system, was the most abundant secreted protein in *Synechococcus* (UniProtKB: A0A4P8WZ44), fourth most abundant in *Synechocystis* (UniProtKB: P73704) and was not identified in *Nostoc*. This was expected as cell surface appendages are amongst the most prominent proteins typically found in the secretomes of unicellular cyanobacteria (Schuergers & Wilde 2015). Despite *Nostoc* possessing a functional T4P system, PilA1 is only expressed in hormogonia (Khayatan, Meeks & Risser 2015; Risser 2023). Given hormogonia were not found during microscopic inspection of the cultures, the absence of PilA1 agrees with the observed lack of differentiated cells (Supplementary Fig. 3).”

Line 458-460 – It would be useful for the authors to clarify the role of hfq here. Hfq directly interacts with PilB and is essential for type IV pilus extension. The phenotypes of hfq mutants in cyanobacteria (loss of motility, competence, aggregation) are due to the loss of T4P function and has important implications for the exoproteome analysis.

We agree and have added some extra text to the manuscript:

“In the Δhfq condition we used a *Synechocystis* Hfq knockout mutant. Current evidence suggests that the cyanobacterial Hfq binds to the T4P extension motor PilB1 and regulates T4P biogenesis. The loss of Hfq, and consequent loss of T4P function, abrogates several processes that depend on pilin proteins such as motility, natural competence and aggregation...”

Line 481-482 – It would be useful to include a list of the proteins that are absent from the hfq mutant exoproteome, I could not seem to find one. Although I would imagine it can be extracted through the available supplemental data, it would be nice to provide this information more directly to the reader. Given that PilA would no longer be exported across the outer membrane in the hfq mutant one would expect PilA to be missing or dramatically reduced in the hfq exoproteome. Was this the case?

Yes, this was the case with PilA1 reduced 59x times in the Hfq mutant exoproteome (average intensity 47539.52 (WT) vs 808.48 (Δhfq)). This has been included in the manuscript.

We have also included a list of the proteins unique to the WT exoproteome (i.e., absent from the Hfq exoproteome) (Supplementary Data 14).

571-573 – Enzymes involved in cell envelope assembly and maintenance would presumably be ubiquitous to all bacteria/living organisms in general. It is not clear why the authors single out cyanobacteria here.

This has been rephrased.

“...enzymes involved in cell envelope assembly and maintenance are ubiquitous and are a promising direction of future studies in cyanobacteria.”

Line 582-583 – Some have speculated that the cyanobacterial T4P system also functions as a T2SS? Would it be worth discussing this in the context of this study? In particular, how does this relate to the data on the hfq mutant, given that mutation of hfq inactivates the T4P system. Again, it would be worth discussing the proteins that are differentially secreted in the hfq mutant vs wild type with this context in mind.

This is again a great point and further investigation led to interesting observations. Please find the details added to the discussion below.

“However, cyanobacteria only possess the homologous T4P system which, some have speculated, could act as a T2SS (Yegorov et al. 2021). Our previous work has shown that inactivation of the T4P OM pore, PilQ, did not change the secretion levels of a heterologous reporter. Thus, suggesting that the T4P system is likely dedicated to the secretion of pilins (Russo et al. 2022). In support of this, PilA1 is significantly less abundant (59-fold) in the secretome of the *Synechocystis* Hfq mutant in this work. Apart from this, virtually all proteins present in both the WT and mutant secretomes are upregulated in the Δhfq condition. A closer look at the 30 proteins absent from the Δhfq secretome also shows that they are all low-abundance proteins (median rank = 636). Taken together, this work supports the conclusion that the T4P system is not a main route for two-step secretion in cyanobacteria.”

REVIEWERS' COMMENTS:

Reviewer #1 (Remarks to the Author):

I think the authors have done a good job responding to mine and the other reviewer's comments. I just have a few minor points:

Given the authors response to my comment about the sll1951 protein I think it is worth the authors spelling out (line 93) the fact that the PCC-M *Synechocystis* strain is motile (and also state that this strain was used because the non-motile lineage of *Synechocystis* has multiple secretion deficiencies (Schuergers and Wilde, 2015).

Line 452: please replace which with what i.e. to investigate to what extent proteins.....

Line 673: number of proteins

Line 733: References contain several with capital letters throughout titles (e.g. Batth, Bruderer, Campbell etc) that require correcting. Also please check species names are in italics (e.g. *Nostoc punctiforme* Line 762)

Reviewer #2 (Remarks to the Author):

I believe the authors have fully addressed my comments and concerns, and so I am happy to recommend this manuscript for publication.

I do have one additional comment - I believe the x axis title in Fig 5C and 5E should contain a fully capitalized 'COG' rather than all lower case 'cog'.

Reviewer #3 (Remarks to the Author):

The revised manuscript sufficiently addresses the concerns raised upon review of the initial manuscript and is now suitable for publication.